# Nup358 restricts ER-mitochondria connectivity by modulating mTORC2/Akt/GSK3β signalling

Misha Kalarikkal, Rimpi Saikia [ID], Lizanne Oliveira [ID], Yashashree Bhorkar, Akshay Lonare [ID], Pallavi Varshney, Prathamesh Dhamale [ID], Amitabha Majumdar [ID] & Jomon Joseph [ID] ✉

## Abstract

ER–mitochondria contact sites (ERMCSs) regulate processes, including calcium homoeostasis, energy metabolism and autophagy. Previously, it was shown that during growth factor signalling, mTORC2/Akt gets recruited to and stabilizes ERMCSs. Independent studies showed that GSK3β, a well-known Akt substrate, reduces ER–mitochondria connectivity by disrupting the VAPB–PTPIP51 tethering complex. However, the mechanisms that regulate ERMCSs are incompletely understood. Here we find that annulate lamellae (AL), relatively unexplored subdomains of ER enriched with a subset of nucleoporins, are present at ERMCSs. Depletion of Nup358, an AL-resident nucleoporin, results in enhanced mTORC2/Akt activation, GSK3β inhibition and increased ERMCSs. Depletion of Rictor, a mTORC2-specific subunit, or exogenous expression of GSK3β, was sufficient to reverse the ERMCS-phenotype in Nup358-deficient cells. We show that growth factor-mediated activation of mTORC2 requires the VAPB–PTPIP51 complex, whereas, Nup358's association with this tether restricts mTORC2/Akt signalling and ER–mitochondria connectivity. Expression of a Nup358 fragment that is sufficient for interaction with the VAPB–PTPIP51 complex suppresses mTORC2/Akt activation and disrupts ERMCSs. Collectively, our study uncovers a novel role for Nup358 in controlling ERMCSs by modulating the mTORC2/Akt/GSK3β axis.

**Keywords** Annulate Lamellae; ER-mitochondria Contact Sites; mTORC2; Nucleoporins; GSK3β
**Subject Categories** Membranes & Trafficking; Organelles; Signal Transduction

## Introduction

Contrary to the general perception that organelles in the cell exist and function independently, recent developments highlight that multiple organelles make physical and dynamic contacts with each other to coordinate the many functions they perform (Henne, 2016; Wu et al, 2018; Prinz et al, 2020; Scorrano et al, 2019). Endoplasmic reticulum (ER), an organelle characterized by its membranous network that extends throughout the cytoplasm, interacts with other organelles through specific contact sites (López-Crisosto et al, 2015; Petkovic et al, 2021; Phillips and Voeltz, 2016; Cohen et al, 2018). ER–mitochondria contact sites (ERMCSs), also known as mitochondria-associated ER membrane (MAM), are maintained through physical interactions between proteins present on both the organelles (Marchi et al, 2014; Raturi and Simmen, 2013; Wu et al, 2018; Rowland and Voeltz, 2012; Filadi et al, 2017). One of the well-characterized tethers is a complex between an ER-resident vesicle-associated membrane protein-associated protein B (VAPB) and a mitochondrial outer membrane protein, protein tyrosine phosphatase-interacting protein 51 (PTPIP51) (Stoica et al, 2014; De vos et al, 2012; Gomez-Suaga et al, 2017; Stoica et al, 2016; Obara et al, 2024). The others include Mfn2-Mfn1/2 (De Brito and Scorrano, 2008), IP3R-GRP75-VDAC (Szabadkai et al, 2006) and BAP31-Fis1 (Iwasawa et al, 2011).

ERMCSs regulate inter-organelle $Ca^{2+}$ transfer, lipid transfer, energy metabolism, inflammation, apoptosis, autophagy and several other ER/mitochondria-dependent functions (Perrone et al, 2020; Barazzuol et al, 2021; Vance, 2020; Csordás et al, 2018; Madreiter-Sokolowski et al, 2019; Giordano, 2018). The ER-resident $Ca^{2+}$ channel IP3R is enriched at the ERMCSs where it mediates $Ca^{2+}$ transfer from the ER to the mitochondria to regulate energy metabolism and apoptosis (Grimm, 2012; Csordás et al, 2018). Although many critical cellular functions are dependent on ERMCSs, the molecular interplay regulating the dynamic interaction between ER and mitochondria remains unclear. Nevertheless, it has been shown that growth factor-stimulated mTORC2 pathway stabilizes ERMCSs through Akt-mediated phosphorylation of ERMCS proteins such as IP3R3 and PACS2 (Betz et al, 2013). Independent reports also show that GSK3β destabilizes ERMCSs by disrupting the VAPB-PTPTIP51 tethering complex (Stoica et al, 2016, 2014; Kors et al, 2022). Growth factor signalling activates mTORC2, which phosphorylates Akt at S473 and completely activates Akt, which has been shown to phosphorylate GSK3β at S9 (Manning and Toker, 2017). However, whether growth factor signalling dependent ERMCS stability involves mTORC2/Akt-mediated inhibition of GSK3β is not known.

The ER is involved in a multitude of physiological functions, which are performed by specialized subdomains of the organelle (Cohen et al, 2018). Annulate lamellae (AL), an underexplored subdomain of the ER, majorly characterized in germ cells, are stacked membranes containing pore-like assemblies that

National Centre for Cell Science, S.P. Pune University Campus, Pune, Maharashtra 411007, India. ✉E-mail: josephj@nccs.res.in

structurally resemble the nuclear pore complexes (NPCs) present on the nuclear envelope (NE) (Kessel, 1992). Although AL are reported in somatic cells, their structural details are unclear. Many functions for AL have been proposed based on electron microscopic observations, which include roles in infection, cancers, gene expression, mRNA regulation and development (Kessel, 1992). AL are remodelled extensively during embryonic development, infection by intracellular pathogens, including SARS-CoV-2 and in cancers (Kessel, 1992; Eymieux et al, 2021). Recently, the involvement of AL in the functional assembly of NPCs has been reported (Hampoelz et al, 2016). AL possess only a subset of nucleoporins as compared to the NPCs (Sahoo et al, 2017; Cordes et al, 1996), thus implying a compositional, structural and functional difference between AL pore complexes and NPCs. A role for the AL-resident nucleoporin Nup358 in microRNA-mediated translational suppression of mRNAs has been documented (Sahoo et al, 2017).

Consistent with the range of processes the ERMCSs regulate, dysfunctional ER–mitochondria crosstalk is implicated in many disorders including diabetes (Rieusset, 2018; Tubbs et al, 2014), cancers (Doghman-Bouguerra and Lalli, 2019; Simoes et al, 2020; Peruzzo et al, 2020) and neurodegeneration (Paillusson et al, 2016; Petkovic et al, 2021). Specifically, ERMCSs and their functions are compromised in neurodegenerative disorders such as amyotrophic lateral sclerosis (ALS)/frontotemporal dementia (FTD), Alzheimer's (AD), Parkinson's (PD) and Huntington's diseases (HD) (Petkovic et al, 2021; Paillusson et al, 2016; Markovinovic et al, 2022). VAPB, an ERMCS tethering protein, is mutated in amyotrophic lateral sclerosis (ALS) (Nishimura et al, 2004). The pathogenic mutation in VAPB, P65S, is shown to stabilise the VAPB–PTPIP51 tethering complex, thus affecting the abundance and functions of ERMCSs (De vos et al, 2012). Impaired ERMCSs are also implicated in the more common forms of ALS/FTD (Fallini et al, 2020). Ectopic expression of the wild-type or ALS/FTD-associated mutant versions of FUS and TDP-43 is shown to decrease ERMCSs through GSK3β-mediated disruption of VAPB–PTPIP51 interaction (Stoica et al, 2014, 2016).

Recently, multiple reports have linked impaired nucleocytoplasmic transport (NCT) to neurodegenerative diseases (Ding and Sepehrimanesh, 2021; Fahrenkrog and Harel, 2018). This may be due to the perturbation of the NPCs, mislocalization and/or degradation of nucleoporins and other components of the NCT (Kim and Taylor, 2017; Spead et al, 2022). However, whether there exists any relationship between the NCT pathway/components and the ERMCS tethering complexes or their functions is unclear.

Since ER makes functional contacts with mitochondria through specialized subdomains at the ERMCSs, and AL represent another subdomain of the ER in the cytoplasm, we explored if these two subdomains associate with each other. We found that Nup358-positive AL are often present at the contact sites between ER and mitochondria. Moreover, Nup358 interacts with the ERMCS tethering complex VAPB–PTPIP51, and loss of Nup358 leads to enhanced ER–mitochondria connectivity through increased mTORC2/Akt pathway activation and GSK3β inhibition. Together, our studies reveal a role for Nup358 in modulating growth factor-mediated remodelling of ERMCSs through a mechanism involving the suppression of mTORC2/Akt pathway and activation of GSK3β.

# Results

## Annulate lamellae are present at ER–mitochondria contact sites

As AL are subdomains of ER, and the ER makes extensive contacts with multiple organelles, including mitochondria, we tested the hypothesis that AL are present at the contacts between ER and mitochondria. Microscopic images of cells co-stained for Nup358 as an AL marker (Sahoo et al, 2017), PDIA3 as an ER marker and MitoTracker for mitochondria revealed that AL ($40 \pm 6\%$; 18 cells) are often present at ERMCSs (Fig. 1A). Consistent with this, super-resolution images obtained by stimulated emission depletion microscopy (STED) of AL (Nup358) and mitochondria (Tom20) showed that many of the AL structures were present adjacent to mitochondria (Fig. 1B). Fractionation of organelles from HeLa cells confirmed that a pool of Nup358, along with other known AL-associated nucleoporins (Nup62 and Nup88) (Sahoo et al, 2017), was present in the mitochondria-associated membrane (MAM) fraction, which represents ERMCSs (Montesinos and Area-Gomez, 2020) (Fig. 1C). In addition, disruption of ERMCSs by depletion of the tethering proteins VAPB/PTPIP51, Mitofusin 2 (Mfn2) or IP3R3, which are essential for ERMCS integrity, led to disappearance of AL, as determined by the absence of cytoplasmic puncta of Nup358 and other AL-associated nucleoporins (Fig. EV1). However, the localization of these nucleoporins to the nuclear membrane remained unaffected under the above condition. Collectively, the results show that AL reside at ERMCSs and the assembly and/or stability of AL depends on ERMCSs.

## Nup358 depletion increases the connectivity between ER and mitochondria

We tested if the AL component Nup358 controls the contacts between the ER and the mitochondria by assessing the ERMCS integrity in Nup358-depleted cells in multiple ways. Knockdown of Nup358 led to increased colocalization between ER and mitochondria in fluorescence microscopy (Fig. 1D). Moreover, siRNA-mediated depletion of Nup358 led to increased interaction between components of the ERMCS tethering complex, VAPB (ER) and PTPIP51 (mitochondria), as monitored by co-immunoprecipitation assays (Fig. 1E). Furthermore, proximity ligation assay (PLA) for two of the ERMCS tethering complexes, VAPB–PTPIP51 and BAP31-Fis1, also indicated that there was increased interaction between the ER and mitochondria (Fig. 1F,G; Appendix Fig. S1). In addition, the relative increase in ERMCSs in Nup358-deficient cells, as compared to control cells, was also evident from the transmission electron microscopy (TEM) studies (Fig. 1H). Collectively, these results suggest that Nup358 depletion led to increased ER–mitochondria connectivity.

Previous studies have shown a role for Nup358 in the miRNA pathway (Shen et al, 2021; Sahoo et al, 2017). However, disruption of the miRNA pathway via depletion of GW182 (Pfaff and Meister, 2013) did not affect ERMCSs (Fig. EV2A), showing that Nup358 modulates ER–mitochondria contacts independent of its function in the miRNA pathway. Moreover, contrary to Nup358 depletion, depletion of Nup214, another AL-resident nucleoporin, led to a decrease in ERMCSs (Fig. EV2B), indicating that different nucleoporins at the AL might have different roles in the regulation of ER–mitochondria contacts. We also found that the absence of Nup358 did not affect AL integrity (Appendix Fig. S2). Depletion of

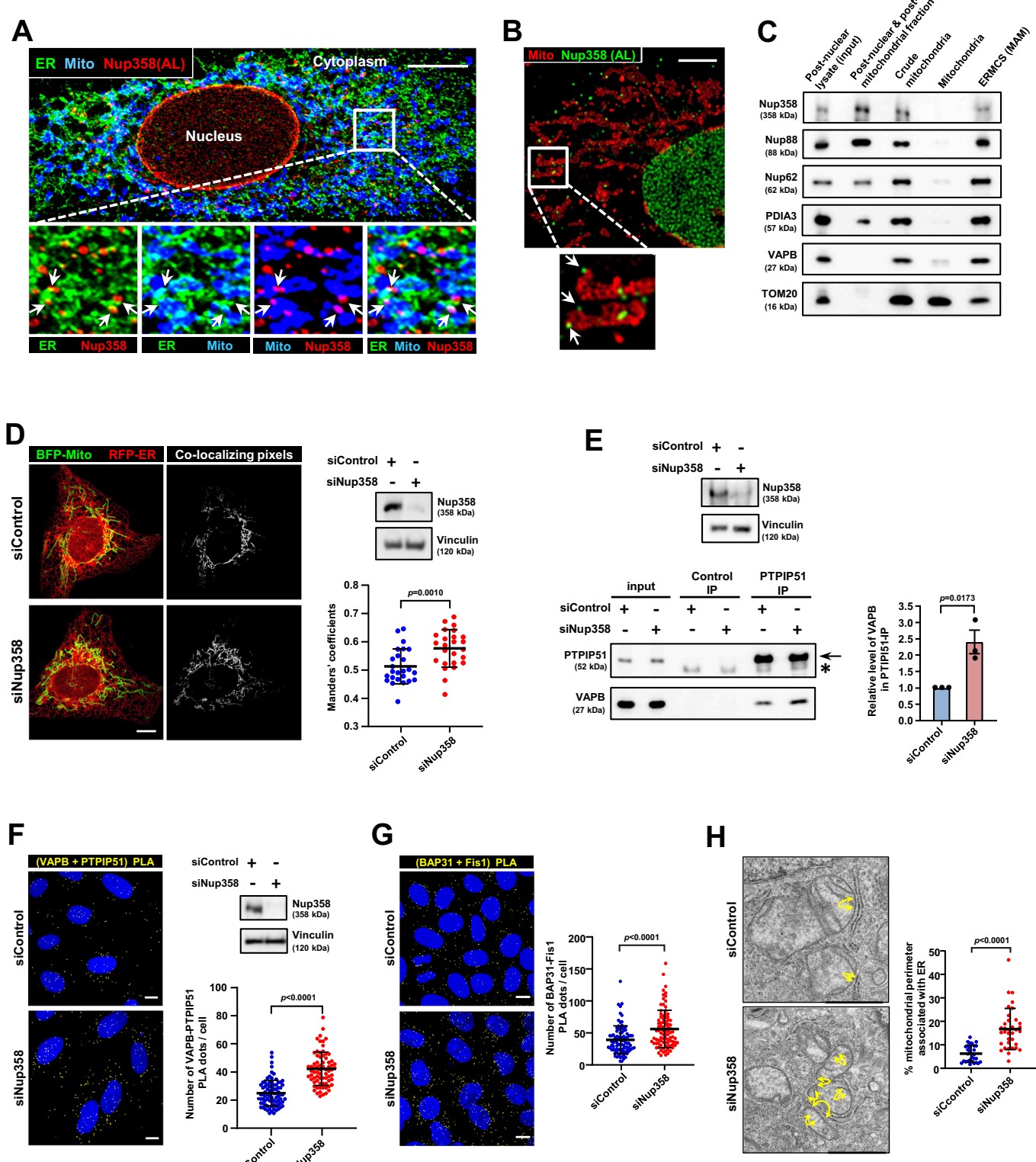

Nup358 by siRNA or CRISPR-mediated knockout (KO, Appendix Fig. S3) did not alter mitochondrial mass or DNA (Appendix Fig. S4), suggesting that the enhanced ER–mitochondria connectivity found in Nup358 depleted cells was not a consequence of increased mitochondrial content.

## Increased ER–mitochondria connectivity in Nup358-deficient cells depends on mTORC2

Growth factor signalling results in the recruitment of mTORC2 to ERMCSs, which in turn increases the ER–mitochondria contacts

Figure 1.   AL-resident Nup358 localizes to and restricts ERMCSs.

(A) Nup358-positive AL are present at ERMCSs. Confocal microscopic image of a U2OS cell displaying the relative localization of indicated proteins; Nup358 for AL (red), PDIA3 for ER (green) and MitoTracker for mitochondria (blue). Arrows show Nup358-positive AL associated with mitochondria. Scale bar, 10 µm. The proportion of Nup358-positive AL associated with mitochondria was 42.8 ± 6.6% (n = 20 cells). (B) Association of Nup358-positive AL with mitochondria. Stimulated emission depletion (STED) super-resolution microscopy of a Huh7 cell immunostained with the AL marker (Nup358, green) and mitochondria marker (TOM20, red). Arrows indicate Nup358-positive AL associated with mitochondria. Scale, bar 5 µm. (C) Nup358, along with other AL-resident nucleoporins like Nup88 and Nup62, is present in the ERMCS fraction. HeLa cells were processed to obtain ERMCS (MAM) fractions, which along with other fractions were analysed for the presence of specific proteins by western blotting. PDIA3, ER marker; TOM20, mitochondrial marker; VAPB, ERMCS marker. (D) Increased ER–mitochondria contacts in Nup358-deficient cells. U2OS cells were initially transfected with control (siControl) or Nup358 (siNup358)-specific siRNA and later co-transfected with RFP-ER (red) and BFP-Mito (pseudo-coloured in green) constructs for labelling ER and mitochondria, respectively. Left: The co-localizing pixels are shown in grey. Scale bar, 10 µm. Right top: The extent of Nup358 depletion was evaluated by western blotting with Nup358-specific antibody. Vinculin was used as a loading control. Right bottom: Analysis of individual Manders' overlap coefficient values of RFP-ER with BFP-Mito from siControl and siNup358-treated HeLa cells (n = 25 cells from three independent experiments). Data are mean ± SD, Student's t test. P value is indicated. (E) Extent of VAPB interacting with PTPIP51 as analysed by co-immunoprecipitation (co-IP) assay. Top: Extent of Nup358 depletion was assessed by western blotting. Bottom left: IP of endogenous PTPIPI51 from HeLa cells treated with siControl or siNup358 was performed and the extent of VAPB co-immunoprecipitated was assessed by western blotting. The arrow indicates PTPIP51 bands and asterisk indicates IgG heavy chain cross-reaction. Bottom right: Quantitation of the amount of VAPB associated with PTPIP51 (n = 3 independent experiments). Data are mean ± SEM, unpaired Student's t test. P value is indicated. (F) Enhanced in situ interaction between VAPB and PTPIP51 in the absence of Nup358. In situ proximity ligation assay (PLA) was performed for assessing the interaction between VAPB and PTIPI51 using specific antibodies. Left: Representative images showing PLA puncta (red) in HeLa cells treated with siControl or siNup358. DNA was stained with Hoechst 33342 (blue). Scale bar, 10 µm. Right top: Extent of Nup358 depletion as analysed by western blotting, along with vinculin as loading control. Right bottom: Quantitation of the number of PLA puncta per cell from siControl and siNup358 HeLa cells (n = 82 cells for siControl and 73 cells for siNup358 from three independent experiments). Data are mean ± SD, Student's t test. P value is indicated. (G) Increased interaction between BAP31 and Fis1 in Nup358-deficient cells. The cells were treated and processed as described in (F). Left: PLA was performed for monitoring the in situ interaction between BAP31 and Fis1, components of another ER–mitochondria tethering complex. Right: Quantitation of the number of PLA puncta per cell from siControl and siNup358 HeLa cells (n = 90 cells for siControl and siNup358 from three independent experiments). Data are mean ± SD, Student's t test. P value is indicated. (H) Increased contacts between ER and mitochondria in Nup358-deficient cells as assessed by transmission electron microscopy (TEM). Left: TEM images displaying ER–mitochondria contacts in the presence (siControl) and absence of Nup358 (siNup358). Contact sites are highlighted with yellow arrows. Scale bar, 1 µm. Right: Quantitative data depicting percentage (%) of mitochondrial surface (perimeter) showing ≤30 nm proximity with the ER membrane (n = 30 mitochondria for siControl and 35 mitochondria for siNup358-treated conditions). Data are mean ± SD, Student's t test. P value is indicated. Source data are available online for this figure.

via Akt-mediated phosphorylation of proteins such as PACS2, hexokinase and IP3R (Betz et al, 2013). Therefore, we examined if the increased ER–mitochondria connectivity that we observed in Nup358-deficient cells depends on mTORC2/Akt. Interestingly, siRNA-mediated depletion of Nup358 from HeLa cells led to the activation of mTORC2, as determined by the phosphorylation of its downstream substrate Akt at S473 (Fig. 2A). Activation of mTORC2 was also confirmed in Nup358 KO HeLa cells (Fig. 2B). Exogenous expression of siRNA-resistant human Nup358 in Nup358-deficient cells rescued the increased activation of mTORC2/Akt (Fig. EV3). Interestingly, depletion of Nup358 in Drosophila also activated mTORC2/Akt pathway, pointing to a conserved role for Nup358 in regulating this pathway (Fig. EV4). Co-depletion of Rictor, a key subunit of mTORC2, reversed the hyper-phosphorylation of Akt resulting from growth factor (insulin and EGF) signalling in Nup358-depleted HeLa cells (Fig. 2C; Appendix Fig. S5). Knockdown of Nup358 also enhanced phosphorylation of Akt at T308 and activated mTORC1, as indicated by the phosphorylation the mTORC1 substrate S6 at S235/S236, which were almost completely reversed by co-depletion of Rictor (Fig. 2D). This suggests that the hyperactivation of mTORC1 in Nup358-deficient cells primarily depends on mTORC2 (Szwed et al, 2021).

As both mTORC1 and mTOCR2 pathways were activated in the absence of Nup358, we tested if the consequent increase in ER–mitochondria contacts in Nup358-deficient cells depended on the mTORC1 and/or mTOC2 pathway. Depletion of Rictor (mTORC2-specific subunit), but not Raptor (mTORC1-specific subunit), reversed the increase in ERMCSs observed in Nup358 knocked down cells (Fig. 2E,F). These results suggest that Nup358 restricts ERMCSs by specifically suppressing mTORC2/Akt activation.

## VAPB–PTPIP51 complex is important for mTORC2/Akt activation upon growth factor stimulation and Nup358 depletion

Previously, it was shown that mTORC2 complex and Akt are recruited to the ERMCSs in response to growth factor signalling (Betz et al, 2013). We confirmed that insulin increased the interaction between VAPB and PTPIP51, as assessed by PLA (Fig. 3A). This increased ER–mitochondria connectivity was mediated by mTORC2, as VAPB–PTPIP51 interactions decreased in the presence of insulin when two of the mTORC2-specific subunits, Rictor and Sin1, were individually depleted (Fig. 3A).

It was already known that ERMCSs are required for proper insulin signalling (Tubbs et al, 2014). To check if specific components at the ERMCSs are involved in the recruitment and/or activation of mTORC2/Akt, different proteins that localize to and regulate ERMCSs such as VAPB/PTPIP51, PACS2, IP3R3 and Mfn2 were depleted in HeLa cells. Interestingly, loss of VAPB and PTPIP51, but not others, significantly attenuated insulin-stimulated mTORC2/Akt activation (Fig. 3B). Under these conditions, the levels of other ERMCS tether proteins remained unchanged (Appendix Fig. S6). Also, the depletion of VAPB and PTPIP51 did not significantly affect the levels of mTORC2 components (Appendix Fig. S7). These results suggested that the VAPB-PTIP51 complex plays a specific role in growth factor-stimulated activation and/or stabilization of activated mTORC2/Akt.

Based on our observation that the ER–mitochondria tethering complex VAPB–PTPIP51 is involved in growth factor-induced mTORC2/Akt activation, we tested if the enhanced mTORC2 activity in the absence of Nup358 was mediated by VAPB–PTPIP51. Interestingly, depletion of VAPB or PTPIP51

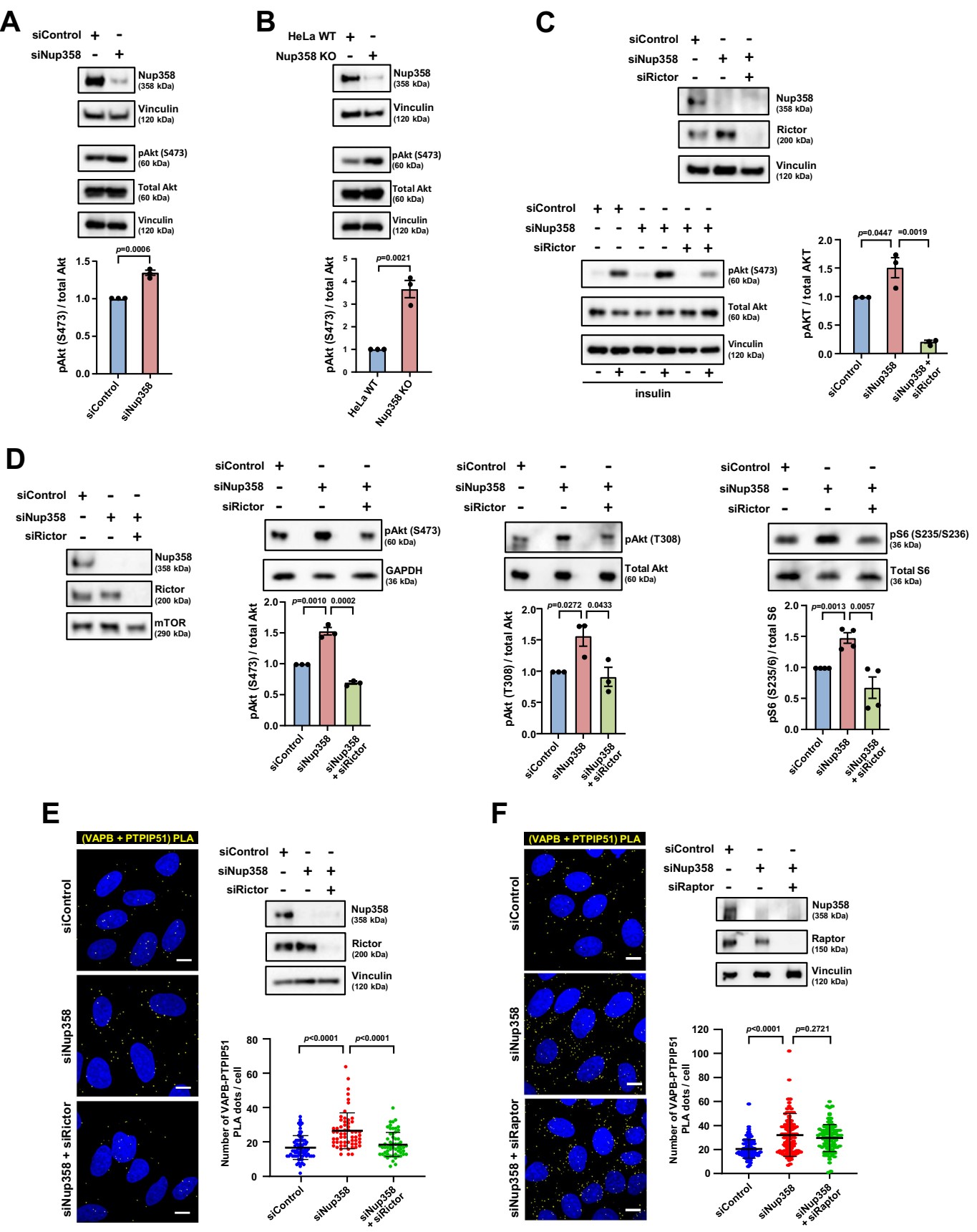

**Figure 2. mTORC2, but not mTORC1, activation is required for enhanced ER–mitochondria contacts in Nup358-downregulated cells.**

(A) Nup358 depletion leads to elevated mTORC2/Akt activation. HeLa cells were treated with Control (siControl) and Nup358 (siNup358) specific siRNA. Top: The cells were then analysed for the extent of mTORC2/Akt activation by western blotting using indicated antibodies. Vinculin was used as a loading control. Bottom: Quantitative representation of the relative levels of pAkt (S473) as compared to total Akt under the above-mentioned experimental conditions ($n = 3$ independent experiments). Data are mean ± SEM, unpaired Student's $t$ test. P value is indicated. (B) Increased activation of mTORC2/Akt signalling occurs in Nup358 knockout (KO) HeLa cells as compared to wild-type (WT) cells. Top: Cells were lysed and subjected to western blotting with indicated antibodies. Bottom: Quantitative data depicting the relative level of pAkt (S473) as compared to total Akt under indicated conditions ($n = 3$ independent experiments). Data are mean ± SEM, unpaired Student's $t$ test. P value is indicated. (C) Nup358 restricts mTORC2-mediated phosphorylation of Akt at S473 upon growth factor signalling. HeLa cells, transfected with indicated siRNAs, were serum starved for 3 h. The cells were then treated with (+) or without (−) insulin (1 nM for 20 min) and analysed for mTORC2/Akt activation by western blotting. Top: The extent of depletion of indicated proteins analysed by western blotting, with vinculin used as a loading control. Bottom left: The extent of Akt phosphorylation at S473 under indicated conditions was determined. Vinculin was used as a loading control. Bottom right: Quantitative representation of the relative levels of pAkt (S473) as compared to total Akt ($n = 3$ independent experiments). Data are mean ± SEM, unpaired Student's $t$ test. P values are indicated. (D) Nup358 depletion leads to mTORC1 activation, which was rescued by co-depletion of Rictor. HeLa cells, treated with indicated specific siRNAs, were analysed for the extent of protein depletion using western blotting, with mTOR being used as a loading control (left). Right top panels: mTORC2 activation was assessed by examining the phosphorylation of Akt at S473. mTORC1 activation was assessed by monitoring the phosphorylation of Akt at T308 and the mTORC1 target S6 at S235 and S236 using western blotting. Lower panels: Quantitation of the relative levels of phosphorylation of specific proteins normalised to GAPDH (for pAkt-S473) or respective total proteins as indicated ($n = 3$ or 4 independent experiments). Data are mean ± SEM, unpaired Student's $t$ test. P values are indicated. (E) Co-depletion of Rictor reverses the increase in ER–mitochondria connectivity in Nup358-depleted cells. HeLa cells were treated with specific siRNAs to deplete indicated proteins and analysed for the extent of ERMCSs present using in situ PLA (yellow) with VAPB and PTPIP51 antibodies (left). DNA was stained with Hoechst 33342 (blue). Scale bar, 10 μm. The extent of protein depletion (right top) and quantitation of PLA dots per cell (right bottom) are shown ($n = 72$ cells for siControl; 59 cells for siNup358; 58 cells for siNup358 + siRictor from three independent experiments). Data are mean ± SD, Student's $t$ test. P values are indicated. (F) The experiment was conducted as described in (E), except that instead of Rictor, as indicated, Raptor-specific siRNA was used. Left: Representative microscopic images displaying the PLA dots (yellow). DNA was stained with Hoechst 33342 (blue). Scale bar, 10 μm. The extent of protein depletion (right top) and quantitation of PLA dots per cell (right bottom) are shown ($n = 90$ cells for siControl; 107 cells for siNup358; 98 cells for siNup358 + siRaptor from three independent experiments). Data are mean ± SD, Student's $t$ test. P values are indicated. Source data are available online for this figure.

rescued the hyperactivation of mTORC2/Akt signalling in Nup358 knockdown cells (Fig. 3C), indicating that Nup358 restricts mTORC2/Akt activity in a VAPB–PTPIP51-dependent manner. This also suggests that assembly of the VAPB–PTPIP51 complex might be essential for mTORC2/Akt activation and/or stabilization of the activated mTORC2/Akt.

## Binding of Nup358 or mTORC2 to VAPB–PTPIP51 complex may determine the extent of ER–mitochondria connectivity

To test if the VAPB–PTPIP51 tethering complex physically interacts with the mTORC2 complex for regulation of the latter, a co-immunoprecipitation assay was performed. Myc-VAPB and HA-PTPIP51 co-immunoprecipitated with FLAG-mTOR (Fig. 4A). Moreover, specific endogenous interaction of mTOR and the mTORC2-specific subunit Rictor with VAPB was confirmed by PLA (Fig. 4A), whereas no interaction between VAPB and the mTORC1-specific subunit Raptor was detected (Fig. EV5). Collectively, the data reveal a specific physical interaction of mTORC2 complex with VAPB, and possibly with the VAPB–PTPIP51 complex.

So far, our data indicate that the ERMCS tethering complex VAPB–PTPIP51 is important for growth factor-induced mTORC2/Akt activation and, thereby, possibly increases the connectivity between ER and mitochondria. The results also suggest that Nup358 restricts ERMCSs by suppressing mTORC2/Akt activation (Fig. 2E). Interestingly, Nup358 also interacts with VAPB and PTPIP51, as determined by co-IP and PLA (Fig. 4B). Given that the associations of Nup358 and mTORC2 with the VAPB–PTPIP51 complex had opposing effects, we hypothesized that perhaps Nup358, when present at the ERMCSs, prevents the association of mTORC2 with the VAPB–PTPIP51 complex, thereby reducing mTORC2/Akt activity and ER–mitochondria connectivity. In other words, Nup358 may inhibit the interaction between mTORC2

complex and VAPB–PTPIP51, in the absence of growth factor signalling. On the contrary, the Nup358-mediated inhibition of mTORC2 association with VAPB–PTPIP51 complex should be relieved in the presence of growth factor signalling, thereby enhancing mTORC2 activity and ER–mitochondria connectivity. To test this possibility, the kinetics of complex formation between Nup358, mTOR, Rictor and Sin1 with VAPB was monitored by PLA in the presence and absence of insulin. Consistent with the hypothesis, as compared to the untreated control condition, insulin treatment resulted in decreased VAPB–Nup358 interaction, but increased VAPB–mTOR, VAPB–Rictor and VAPB–Sin1 association (Fig. 4C). These results indicated a counteracting role for Nup358 and mTORC2 in regulating the extent of ER–mitochondria connectivity during growth factor signalling, possibly mediated by their mutually exclusive interaction with the VAPB–PTPIP51 tethering complex at ERMCSs.

In an attempt to understand how insulin mediates the decreased Nup358–VAPB interaction, we explored a role for PI3 kinase—a well-known mediator of insulin/growth factor signalling (Saxton and Sabatini, 2017). Towards this, cells were treated with insulin in the presence or absence of wortmannin, a PI3 kinase-specific inhibitor, and the interaction between Nup358 and VAPB was assessed using PLA. Insulin decreased Nup358–VAPB interaction as compared to that in untreated control; however, wortmannin did not have any effect on insulin's ability to decrease the interaction (Fig. 4D). These results suggest that Nup358–VAPB interaction may be modulated by growth factor signalling in a PI3 kinase-independent manner.

## Ectopic expression of GSK3β rescues increased ER–mitochondria connectivity caused due to Nup358 depletion

Growth factor signalling has been shown to inhibit GSK3β (Cohen and Frame, 2001). GSK3β independently has been reported

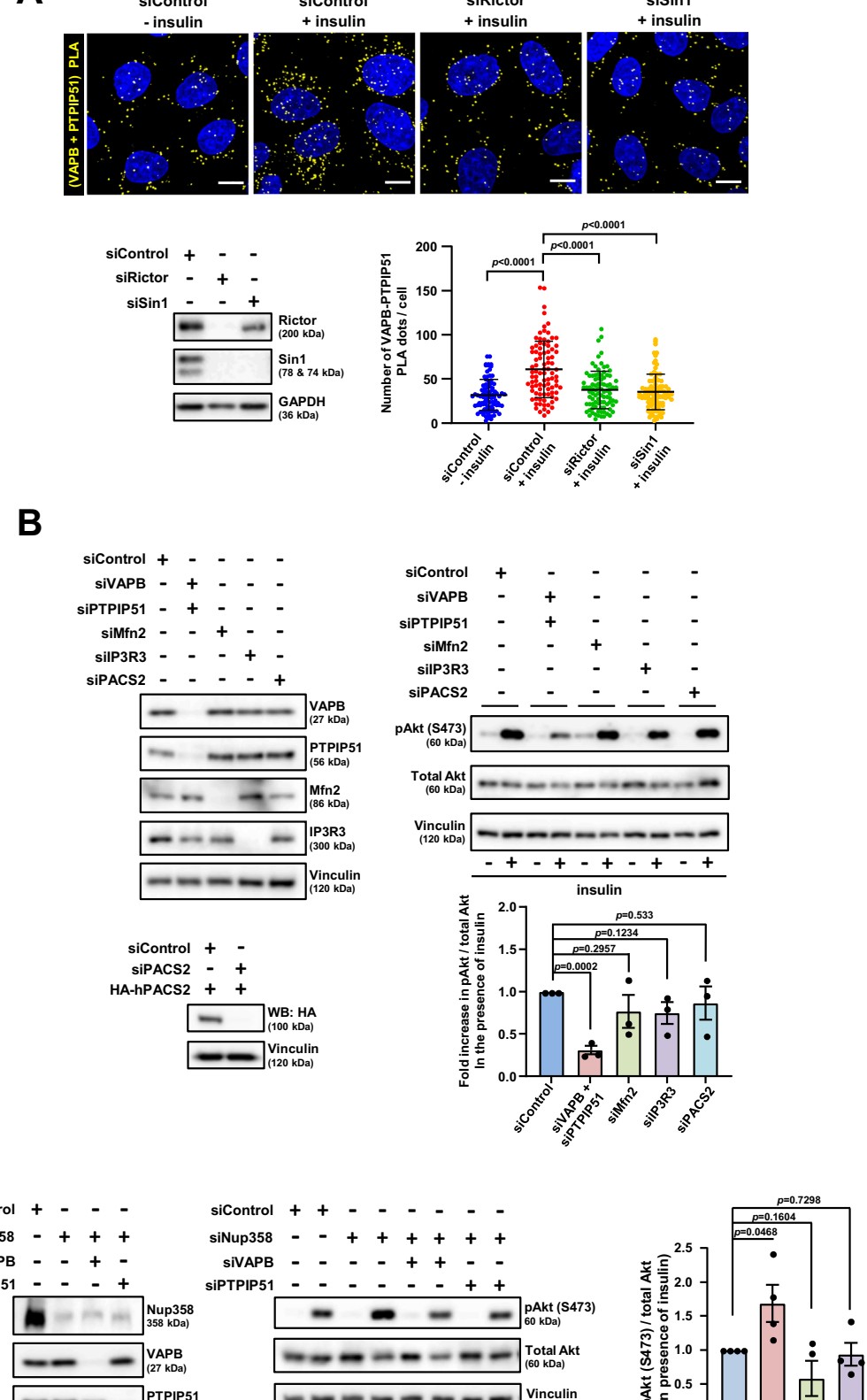

**Figure 3.  VAPB–PTPIP51 tethering complex contributes to increased mTORC2 activation mediated by insulin signalling and caused by Nup358 depletion.**

(A) Insulin-dependent increase in VAPB–PTPIP51 interaction is dependent on mTORC2. HeLa cells were transfected with control (siControl), Rictor (siRictor) or Sin1 (siSin1). Cells were later serum starved for 12 h and treated with (+) or without (−) 2 nM insulin for 20 min. Top: Cells were fixed and processed for PLA using VAPB and PTPIP51-specific antibodies (yellow dots). DNA was stained with Hoechst 33342 (blue). Scale bar, 10 µm. Bottom left: Depletion of Rictor and Sin1 was monitored by western blotting using specific antibodies GAPDH was used as loading control. Note that Rictor depletion led to co-depletion of Sin1 as reported earlier (Yang et al, 2006). Bottom right: Quantitation of PLA dots per cell are shown (n = 90 cells for siControl, siRictor or siSin1 condition from three independent experiments). Data are mean ± SD, Student's *t* test. *P* values are indicated. (B) Depletion of VAPB/PTPIP51 interferes with growth factor-stimulated activation of mTORC2/Akt signalling. HeLa cells were depleted of different proteins using specific siRNAs as mentioned. Left top: The extent of depletion was verified by western blotting using specific antibodies. Left bottom: The ability of PACS2 siRNA to deplete the human PACS2 was confirmed by co-transfecting HeLa cells with control (siControl) or PACS2 (siPACS2) siRNA along with HA-human PACS2 construct. The expression of HA-PACS2 was monitored by western blotting (WB) with a HA-specific antibody. Vinculin was used as a loading control. Right top: HeLa cells depleted of indicated proteins were serum starved for 3 h, and later treated with (+) or without (−) insulin (1 nM) for 20 min. The cells were then analysed for the specific proteins by western blotting. Right bottom: Quantitative analysis of the relative phosphorylation of Akt at S473 under the described conditions (n = three independent experiments). Data are mean ± SEM, unpaired Student's *t* test. *P* values are indicated. (C) Hyperactivation of mTORC2/Akt signalling upon Nup358 loss depends on the VAPB–PTPIP51 tethering complex. HeLa cells were depleted of indicated proteins and serum starved for 3 h. The cells were then treated with (+) or without (−) insulin (1 nM) for 20 min. Cells were analysed for insulin-induced mTORC2/Akt activation. Left: Western analysis to monitor the extent of protein depletion by siRNAs. Middle: Western blot showing the extent of phosphorylation of Akt at Ser473, along with total Akt levels, in the described conditions. Vinculin was used as a loading control. Right: Quantitative data depicting the change in a relative amount of pAkt (S473) under the indicated conditions (n = 4 independent experiments). Data are mean ± SEM, unpaired Student's *t* test. *P* values are indicated. Source data are available online for this figure.

to reduce ERMCSs via disruption of the interaction between VAPB and PTPIP51 (Stoica et al, 2016, 2014). We found that Nup358 depletion led to inhibition of GSK3β, evident by the increase in phosphorylation of GSK3β at S9 as compared to control cells, which could be reversed by co-depletion of Rictor (Fig. 5A). Further we assessed if this inhibition of GSK3β activity is responsible for increased connectivity between ER and mitochondria. Nup358 depletion from GFP-control expressing cells showed increased ERMCSs compared to control siRNA-treated cells. Interestingly, GSK3β -GFP expression reversed the increase in the ER–mitochondria connectivity that was observed in GFP-control expressing Nup358-deficient cells (Fig. 5B). Moreover, co-immunoprecipitation and proximity ligation assays revealed that Nup358 specifically interacts with GSK3β (Fig. 5C). Collectively, these results support the conclusion that in Nup358-deficient cells increased ER–mitochondria connectivity primarily results from decreased GSK3β activity.

## Insulin-dependent remodelling of ERMCSs is abrogated in Nup358-deficient cells

Based on the results so far, it appears that Nup358, possibly along with GSK3β, attenuates the growth factor-dependent increase in ER–mitochondrial connectivity. This requires Nup358-dependent restriction of the mTORC2 activity, maybe at the ERMCS, which could lead to increased GSK3β activity, thereby decreasing ERMCSs. This predicts that in the absence of Nup358, insulin-dependent stabilization of ERMCSs could be significantly more than that in the control cells. To test this, insulin-dependent remodelling of ERMCSs was monitored in control and Nup358 siRNA-treated cells. In control siRNA-transfected cells, insulin treatment led to a significant increase in the ERMCSs (Fig. 5D). In Nup358-depleted cells, even in the absence of insulin, ERMCSs were increased as compared to siControl cells without insulin treatment (Fig. 5D). Interestingly, insulin did not further enhance the contacts between ER and mitochondria in Nup358-depleted cells, as compared to insulin-untreated Nup358-deficient cells (Fig. 5D). These results showed that absence of Nup358 leads to increase in the ERMCSs in an insulin-independent manner, indicating that growth factor signalling mediated removal of Nup358 possibly from the contact sites is sufficient to achieve increased ERMCSs.

## A region of Nup358 encompassing 1949–2786 amino acid residues is sufficient to interact with VAPB/ PTPIP51, restrict mTORC2 activity and reduce ERMCSs

Nup358 is a large nucleoporin with multiple domains (Fig. 6A). Different fragments of Nup358 were assessed for their ability to interact with VAPB/PTPIP51. Co-immunoprecipitation assays showed that a fragment of Nup358—Nup358-MC1 (1949–2786 aa)—possessing two Ran-binding domains (RBD2 and RBD3), a kinesin/dynein-binding domain (KBD/DBD) and internal repeats (IR), was sufficient to interact with VAPB and PTPIP51 (Fig. 6A,B). In vitro interaction studies using purified recombinant proteins indicated that PTPIP51 directly interacts with Nup358-MC2 (2012–2771 aa) (Fig. 6C). Moreover, ectopic expression of Nup358-MC1 resulted in attenuation of mTORC2-dependent Akt phosphorylation as compared to GFP-control expressing cells (Fig. 6D). In addition, expression of Nup358-MC1 was sufficient to rescue the increased mTORC2/Akt activation caused due to Nup358 depletion (Fig. 6E). Nup358-MC1 expression also decreased the abundance of ERMCSs in cells as assessed by VAPB–PTPIP51 PLA (Fig. 6F). Collectively, these results suggest that Nup358-MC1 is sufficient to interact with VAPB–PTPIP51 and restrict mTORC2/Akt-dependent enhancement of ERMCSs.

Based on our study, we propose the following working model (Fig. 7). When growth factors are absent, interaction of Nup358 with VAPB–PTPIP51 complex restricts mTORC2/Akt access to the ERMCS tether, thus activating GSK3β, leading to destabilization of the ER–mitochondria contacts. Upon activation of growth factor signalling, through an unknown mechanism, Nup358 may be removed from the ERMCS, allowing mTORC2/Akt to get recruited to the preformed VAPB–PTPIP51 complex, and further stabilize the ERMCS though a mechanism involving inhibitory phosphorylation of GSK3β. In other words, fine tuning of ERMCSs is achieved by a reciprocal binding of Nup358 or mTORC2 complex to the VAPB–PTPIP51 tethering complex and regulating GSK3β-dependent destabilization of the VAPB–PTPIP51 tethering complex.

## Discussion

Our studies provide new insights into the potential functions of the underexplored organelle AL. We find that Nup358-positive AL are

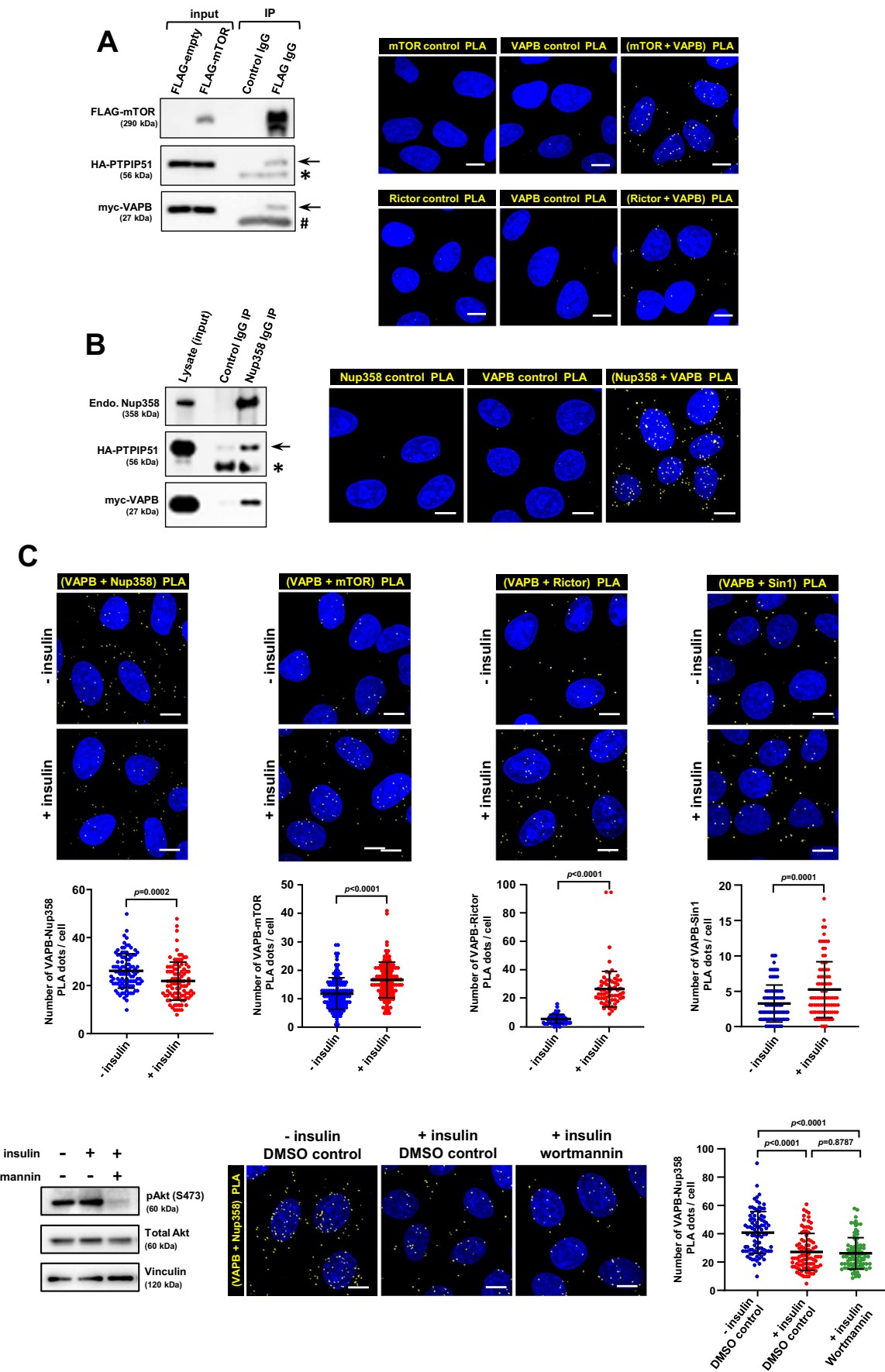

**Figure 4. Growth factor signalling modulates reciprocal binding of mTORC2 and Nup358 to VAPB–PTPIP51 complex.**

(A) mTORC2 interacts with VAPB and PTPIP51. Left: HEK293T cells expressing FLAG-control (Empty vector) or FLAG-mTOR along with HA-PTPIP51 and myc-VAPB were subjected to immunoprecipitation (IP) using FLAG-specific antibodies. Presence of HA-PTPIP51 and myc-VAPB in the IP samples was examined by western analysis. Arrow indicates HA-PTPIP51 or myc-VAPB as shown; * indicates IgG heavy chain and # indicates IgG light chain cross-reaction. Right: In situ interaction of VAPB with mTOR and Rictor. Interaction between endogenous VAPB with mTOR (top) and the mTORC2-specific subunit Rictor (bottom) was confirmed by in situ PLA in HeLa cells. An increase in the number of PLA dots (yellow) was observed with mTOR or Rictor in combination with VAPB antibodies, as compared to single-antibody controls (mTOR, Rictor or VAPB alone). DNA was stained with Hoechst 33342 (blue). Scale bar, 10 μm. (B) Nup358 interacts with PTPIP51 and VAPB. Left: HEK293T cells expressing HA-PTPIP51 and myc-VAPB were subjected to endogenous Nup358 IP (Nup358 IgG IP) using specific antibodies. Presence of HA-PTPIP51 and myc-VAPB was examined by western analysis of the IP samples. Rabbit IgG (Control IgG IP) was used as control. Arrow indicates HA-PTPIP51 and * indicates IgG heavy chain cross-reaction. Right: Interaction between endogenous Nup358 and VAPB was confirmed by in situ PLA using specific antibodies in HeLa cells. Nup358 or VAPB antibody alone was used as control. DNA was stained with Hoechst 33342 (blue). Scale bar, 10 μm. (C) Reciprocal binding of Nup358 and mTORC2 to VAPB is modulated by growth factor signalling. HeLa cells were serum starved for 3 h (12 h for VAPB–Sin1 PLA experiment). Cells were later treated with (+) or without (−) insulin (1–2 nM) for 20 min. Representative images showing the extent of the interaction between Nup358 and VAPB (top first), mTOR and VAPB (top second), Rictor and VAPB (top third) or Sin1 and VAPB (top fourth) as assessed by PLA (yellow dots). DNA was stained with Hoechst 33342 (blue). Scale bar, 10 μm. Bottom panels: Quantitative data depicting the number of PLA dots per cell under the indicated scenario. Number of Nup358–VAPB PLA dots per cell ($n = 98$ cells for insulin-untreated and $n = 97$ for insulin-treated condition from three independent experiments), number of mTOR-VAPB PLA dots per cell ($n = 131$ cells for insulin-untreated and 125 cells for the insulin-treated condition from three independent experiments), number of Rictor-VAPB PLA dots per cell is shown ($n = 67$ cells for insulin-untreated and 61 cells for insulin-treated condition from three independent experiments) and Sin1-VAPB PLA dots per cell ($n = 90$ cells, from three independent experiments) is shown. Data are mean ± SD, Student's $t$ test. $P$ values are indicated. (D) PI3 kinase activity is not required for insulin-dependent decrease in the interaction between Nup358 and VAPB. As indicated, HeLa cells were serum starved for 12 h, post which they were treated without (−) or with (+) the PI3 kinase inhibitor wortmannin (2.5 μM). The cells were then left untreated (−) or treated with 2 nM insulin for 20 min. Left: Cells were lysed and analysed for different proteins as shown. Middle: Cells were subjected to PLA for monitoring the in situ interaction between Nup358 and VAPB (yellow dots). Scale bar, 10 μm. Right: Quantitation of PLA dots per cell for different conditions is shown ($n = 90$ cells for each indicated condition from three independent experiments). Data are mean ± SD, Student's $t$ test. $P$ values are indicated. Source data are available online for this figure.

often present at the contact sites between ER and mitochondria, which are known to play a pivotal role in cellular homoeostasis by regulating critical processes such as inter-organelle transfer of $Ca^{2+}$ and lipids, mitochondrial energetics, cellular metabolism, apoptosis and autophagy. Depletion of Nup358 increased the contacts between ER and mitochondria via enhanced activation of mTORC2/Akt and inhibition of the Akt substrate GSK3β. Thus, the findings of our study highlight the importance of the AL-resident nucleoporin Nup358 in modulating the ER–mitochondria connectivity.

How does Nup358, which interacts with both VAPB and PTPI51, eventually disrupt the VAPB–PTPIP51 interaction to decrease the ER–mitochondria connectivity? One possibility is that Nup358 somehow increases the accessibility and/or activity of GSK3β, which is known to disrupt VAPB–PTPIP51 interactions (Stoica et al, 2014, 2016). The other possibility is that Nup358 may make independent complexes with VAPB and PTPIP51 to achieve the disruption of ERMCSs. Understanding the molecular details requires further investigation.

While investigating the role of Nup358-positive AL in cellular functions, we also uncovered a mechanism by which ERMCS components regulate mTORC2/Akt activation in response to growth factor signalling. Earlier studies have shown that growth factors can stimulate the recruitment of mTORC2/Akt to ERMCSs, leading to Akt-mediated phosphorylation of proteins at the contacts, thereby regulating the dynamics and functions of ERMCSs (Betz et al, 2013). The mTORC2/Akt complex has been shown to be present at multiple intracellular locations, but the site and mechanism of its activation upon growth factor signalling remain undetermined (Ebner et al, 2017; Fu and Hall, 2020). Our studies indicate that the ERMCS tethering complex VAPB–PTPIP51 is important for the activation of the mTORC2/Akt pathway, possibly via its recruitment to the ERMCSs. This is consistent with an earlier report that illustrated the significance of the ERMCS integrity for insulin signalling (Tubbs et al, 2014; Tubbs and Rieusset, 2017). However, we cannot rule out the

possibility that mTORC2 activation could occur elsewhere (for example, at the plasma membrane or endosomes (Fu and Hall, 2020; Knudsen et al, 2020; Szwed et al, 2021)), and the VAPB–PTIPI51 complex may be required to stabilize the activated mTORC2/Akt at the ERMCSs. Nevertheless, our study highlights the importance of VAPB–PTPIP51 tethering complex in regulating the growth factor-dependent activity of mTORC2.

Our results suggest a molecular interplay between Nup358 and mTORC2/Akt at the ERMCSs for regulating ERMCS dynamics. A mutually exclusive binding of Nup358 and mTORC2 complex to the ERMCS tethering complex VAPB–PTPIP51 appears to regulate the ERMCSs, and, perhaps their functions, in response to growth factors. Localization of Nup358 to the ERMCSs, as indicated by the interaction between Nup358 and VAPB, was dependent on insulin signalling. Interestingly, this was independent of the PI3 kinase activity. Currently, the mechanistic details of how Nup358–VAPB interaction is reduced upon insulin signalling remain unresolved.

It is well-established that growth factor signalling can activate Akt, which in turn can inactivate GSK3β through inhibitory phosphorylation of the S9 residue (Cohen and Frame, 2001; Frame and Cohen, 2001). Earlier, it has been shown that GSK3β destabilizes the ERMCSs by disrupting the VAPB–PTPIP51 interaction (Stoica et al, 2016; Paillusson et al, 2016; Stoica et al, 2014). Independently, growth factor signalling, through the mTORC2/Akt activity, is shown to stabilize ERMCSs (Betz et al, 2013). Our results suggest that one of the main mechanisms by which the growth factor-induced stabilization of ERMCSs is achieved is through mTORC2/Akt-mediated suppression of GSK3β.

How is Nup358 function at the ERMCSs regulated? Interestingly, in Nup358-depleted cells, ER–mitochondria connectivity is significantly increased even in the absence of insulin signalling. Moreover, insulin failed to enhance the ERMCSs further in Nup358-deficient cells. This indicates that one of the mechanisms by which growth factors may stabilize ERMCSs is through the removal of Nup358 from the contacts. This allows mTORC2/Akt

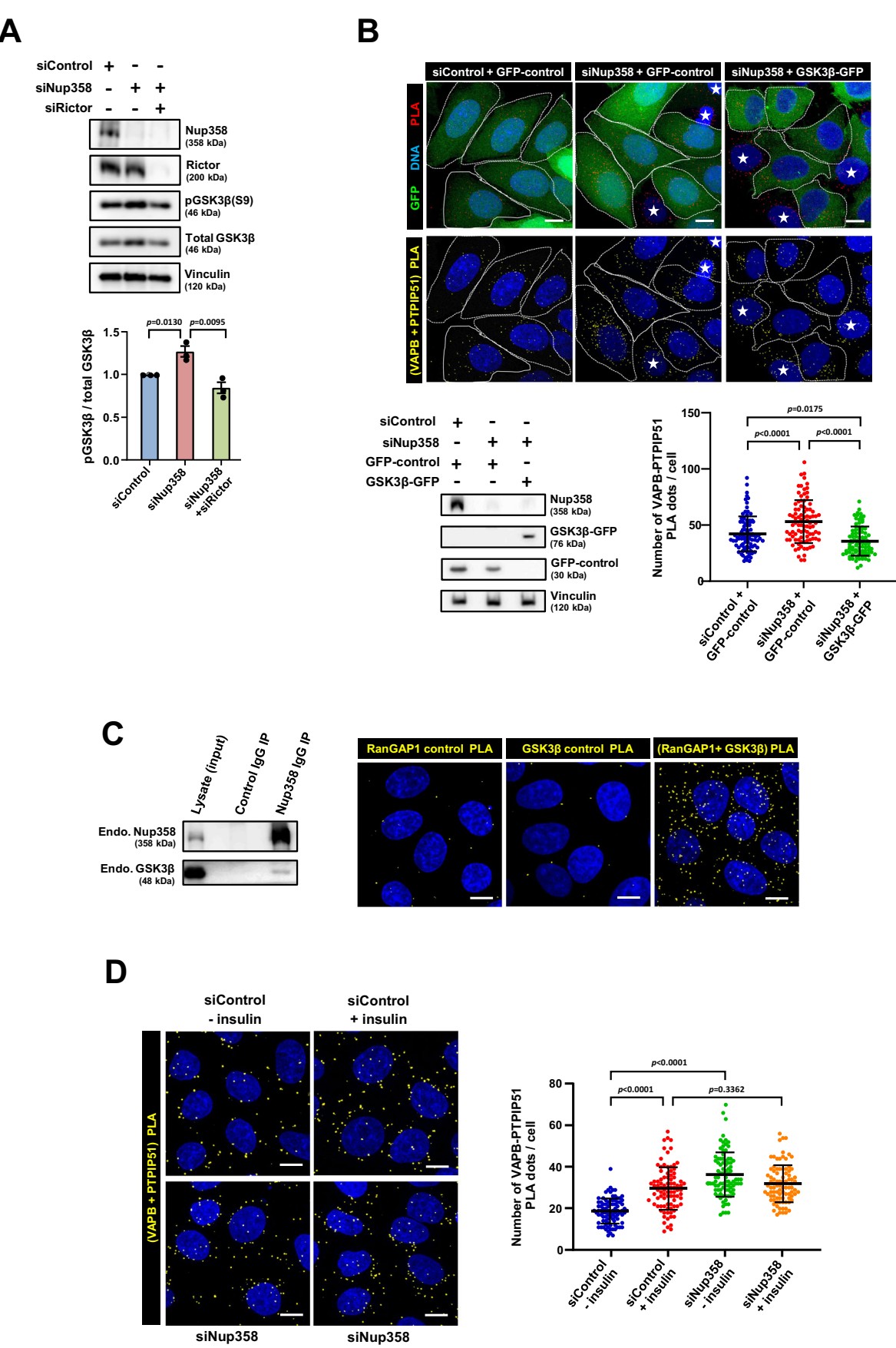

◀ **Figure 5. Inhibition of GSK3β is required for increase in ER–mitochondria contacts upon Nup358 depletion.**

(A) Nup358 depletion leads to increased inhibitory phosphorylation of GSK3β (S9), which could be reversed by co-depletion of the mTORC2-specific subunit Rictor. Cells were transfected with siControl, siNup358 or siNup358 along with siRictor. Top: Cells were lysed and checked for the depletion and the relative levels of phospho-GSK3β (S9) as compared to total GSK3β and Vinculin (loading control) by western blotting. Bottom: Quantitative data depicting the change in a relative amount of pGSK3β (S9) under the indicated conditions ($n = 3$ independent experiments). Data are mean ± SEM, unpaired Student's $t$ test. $P$ values are indicated. (B) Ectopic expression of GSK3β reverses the enhanced ER–mitochondria connectivity found in Nup358-depleted cells. Stable tetracycline-inducible HeLa cell lines expressing GFP-control or GSK3β-GFP were transfected with either control (siControl) or Nup358-specific (siNup358) siRNAs as depicted. The medium was then replaced with 10% FBS-containing medium with doxycycline to induce GFP or GSK3β-GFP for 24 h. Top: PLA was performed for examining the interaction between VAPB and PTPIP51 (yellow dots). DNA was stained with Hoechst 33342 (blue). Scale bar, 10 μm. *Indicates low on no GFP-expressing cells. Cell boundaries are demarcated with dotted lines. Bottom left: Nup358 depletion and GFP-control or GSK3β-GFP expression was monitored by western blotting. Bottom right: Quantitation of PLA dots per cell (GFP-positive) for different indicated conditions is shown ($n = 90$ cells for each indicated condition from three independent experiments). Data are mean ± SD, Student's $t$ test. $P$ values are indicated. (C) Nup358 interacts with GSK3β. Left: Endogenous (Endo.) Nup358 was immunoprecipitated (IP) from HEK293T cells using specific antibodies (Nup358 IgG IP) and Rabbit IgG was used as a negative control. The IP samples were analysed for the presence of indicated proteins by western blotting. Right: Interaction between endogenous RanGAP1 and GSK3β was examined by in situ PLA using specific antibodies in HeLa cells. Here, RanGAP1 was used as a surrogate for Nup358, as RanGAP1 is known as a strong binding partner that colocalizes with Nup358 at the nuclear envelope and AL (Joseph et al, 2004; Sahoo et al, 2017). RanGAP1 or GSK3β antibody alone was used as control. PLA dots are indicated in yellow colour. DNA was stained with Hoechst 33342 (blue). Scale bar, 10 μm. (D) Nup358-deficient cells show increased ER–mitochondria contacts even in the absence of insulin. HeLa cells transfected with siControl or siNup358 for 60 h were serum starved for 12 h, post which they were treated with (+) or without (−) 2 nM insulin for 20 min. Left: Cells were processed for PLA to monitor the interaction between VAPB and PTPIP51 (yellow dots). Right: Quantitation of PLA dots per cell is shown [$n = 90$ cells for siControl (− insulin), siControl (+ insulin), siNup358 (− insulin) or siNup358 (+ insulin) from three independent experiments]. Data are mean ± SD, Student's $t$ test. $P$ values are indicated. Source data are available online for this figure.

recruitment to the ERMCS and inhibition of GSK3β by Akt-mediated S9 phosphorylation. Our study also shows that GSK3β functions downstream to Nup358, as ectopic expression of GSK3β was sufficient to negate the increased ERMCSs in Nup358-depleted cells. Nup358 interacts with GSK3β and may be required for its recruitment to and/or activation at the ERMCSs. Nup358 depletion may result in almost complete inactivation of GSK3β and consequently a maximum increase in the ERMCSs, and therefore, insulin may not further stabilize ER–mitochondria connectivity in the absence of Nup358. Understanding the mechanistic details of how the molecular interplay between VAPB–PTPIP51 complex, mTORC2/Akt, Nup358 and GSK3β are functionally linked with ERMCS remodelling requires future investigation.

In light of the new findings, it is unclear why AL components such as Nup358 are involved in regulating cellular signalling and cytoplasmic processes. It is possible that there exists a crosstalk between nucleocytoplasmic transport (NCT) and cytoplasmic signalling at the ERMCSs. Based on the ultra-structural studies, an interconnection between the nuclear envelope and AL has been proposed (Kessel, 1992). Moreover, the budding of nucleoporin-containing structures and fusion of them with pre-existing AL in the cytoplasm have been observed (Sahoo et al, 2017). The possible interplay between NCT and AL is particularly interesting in light of the implications of dysregulated NCT in the pathogenesis of neurodegenerative diseases (Ding and Sepehrimanesh, 2021). The newly identified role of AL-resident nucleoporins in controlling events other than NCT might be relevant in understanding the pathophysiology of neurodegenerative diseases.

From this study, it is evident that Nup358 and GSK3β function together to destabilize the ERMCSs. Impaired ER–mitochondria connection and its functions are implicated in a wide range of neurodegenerative diseases (Markovinovic et al, 2022). Moreover, increased GSK3β activity appears to be a common feature of most of these diseases (Wang et al, 2023). GSK3β has been shown to be activated upon expression of ALS/FTD-linked genes TDP-43 and FUS, leading to decreased ER–mitochondria connectivity and a concomitant decrease in the levels of mitochondrial $Ca^{2+}$ (Stoica et al, 2014, 2016), potentially altering the mitochondrial bioenergetics. Interestingly, mice lacking Nup358 expression in Thy1$^+$-

motor neurons develop ALS-like symptoms (Cho et al, 2017). The finding that Nup358 regulates the ERMCSs through GSK3β opens up avenues to look into the possible mechanisms of how nucleoporin dysregulations impinge on multiple neurodegenerative diseases.

Nup358 mutations are linked with acute necrotizing encephalopathy 1 (ANE-1), a disease condition triggered by viral infection, and characterized by lesions in specific regions within the brain of affected individuals (Neilson et al, 2009; Neilson, 2010). The pathophysiology includes cytokine storm, exemplified by increased levels of pro-inflammatory cytokines such as INF-α and IL-6 in the serum and cerebrospinal fluid of the patients (Levine et al, 2020). Impairment in the miRNA-mediated translation regulation by Nup358 might contribute to the development of ANE-1 (Deshmukh et al, 2021; Shen et al, 2021). However, dysregulation of the functions identified from our studies for the AL-associated Nup358 may provide an additional/alternative mechanism for the pathogenesis of ANE-1.

In summary, our results suggest that AL structures are often present at the ERMCSs, and one of the AL-resident nucleoporins, Nup358, can restrict ERMCSs via mTORC2/Akt inhibition and GSK3β activation. Our findings also highlight a role for the ERMCS tethering complex— VAPB–PTPIP51—in the recruitment and activation of mTORC2/Akt in response to growth factors. Alternatively, mTORC2/Akt that is activated elsewhere in response to growth factor signalling may be recruited to and stabilized at the VAPB–PTPIP51 complex. Irrespective of the site of activation, it appears that the association of an active mTORC2 complex with the ERMCS tether VAPB–PTPIP51 further increases the contacts between ER and mitochondria possibly via phosphorylation of contact site proteins (Betz et al, 2013) and/or inhibiting GSK3β-mediated disruption of VAPB–PTPIP51 tethering complex (Stoica et al, 2016, 2014; Paillusson et al, 2016). Our study thus reveals a hitherto unidentified role for the AL-associated nucleoporin Nup358, in the regulation of growth factor-mediated ERMCS remodelling as described above. Further studies may shed light on the potential implication of this finding in understanding the mechanisms involved in neurodegenerative diseases and ANE-1.

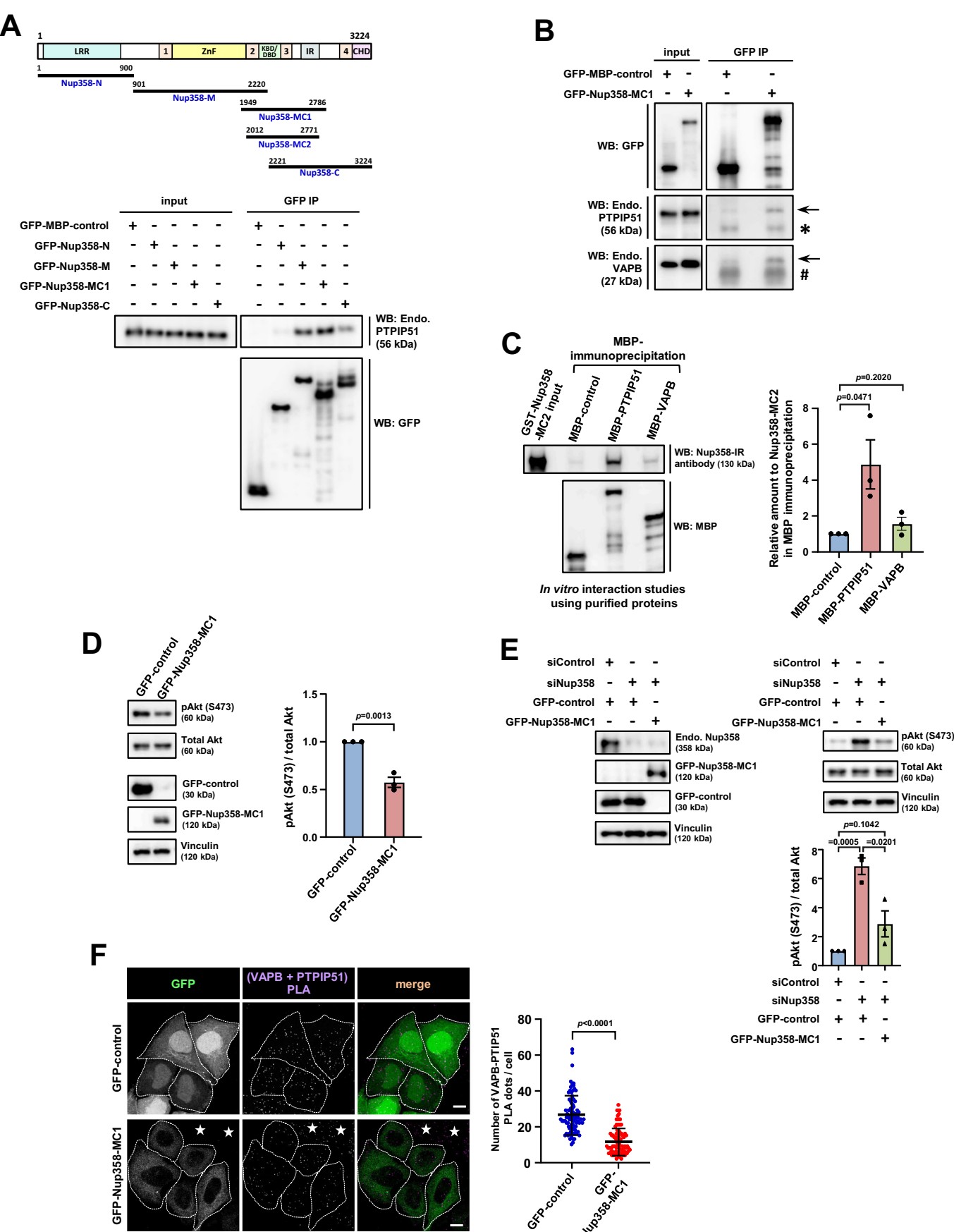

◀  **Figure 6.  A region of Nup358 encompassing 1949 to 2786 amino acid residues is sufficient to restrict mTORC2 activity and ERMCSs.**

(A) A fragment of Nup358 encompassing RBD2, KBD/DBD, RBD3 and IR (Nup358-MC1; 1949–2786 amino acid residues) is sufficient to interact with PTPIP51. Domain architecture and fragments of Nup358 used in this study. LRR leucine-rich region, 1,2,3,4 RanGTP-binding domains (RBDs), ZnF zinc-finger domains, KBD/DBD kinesin-binding domain/dynein-binding domain, IR internal repeats, CHD cyclophilin-homology domain. Amino acid residues are numbered. Bottom: HEK293T cells expressing GFP-MBP-control or GFP-tagged version of different Nup358 fragments as indicated were subjected to IP using GFP-specific antibodies. The IP samples were probed for the presence of endogenous PTPIP51 by western blotting. (B) Nup358-MC1 is sufficient to interact with both VAPB and PTPIP51. HEK293T cells expressing GFP-MPB-control or GFP-Nup358-MC1 were subjected to IP using GFP-specific antibodies. The IP samples were probed for the presence of endogenous PTPIP51 and VAPB by western blotting. Arrow indicates the bands corresponding to PTPIP51 and VAPB. * and # indicate bands corresponding to antibody heavy and light chains, respectively. (C) Nup358-MC2 directly interacts with PTPIP51. Purified GST-tagged Nup358-MC2 (2012 to 2771 amino acid residues) was incubated with MBP-control, MBP-PTPIP51 or MBP-VAPB. The MBP proteins were immunoprecipitated with specific antibodies. Left: The IP samples were analysed for the presence of Nup358-MC2 by western blotting (WB) Nup358-MC2 was detected using an antibody raised against the IR region (Joseph et al, 2004). Right: Quantitative analysis showing the relative levels of Nup358-MC2 detected in the indicated MBP IP samples ($n =$ three independent experiments). Data are mean ± SEM, Student's *t* test. *P* values are indicated. (D) Nup358-MC1 is sufficient to restrict mTORC2/Akt signalling. Left: HeLa cells stably expressing inducible GFP-control or GFP-Nup358-MC1 were induced with doxycycline, lysed and subjected to western blotting using indicated antibodies. Right: Quantitative representation depicting the relative levels of pAkt normalized to total Akt ($n = 3$ independent experiments). Data are mean ± SEM, Student's *t* test. *P* values are indicated. (E) Nup358-MC1 expression is sufficient to rescue increased activation of mTORC2/Akt signalling in Nup358-deficient cells. siControl or siNup358 were initially transfected into HeLa cells expressing inducible GFP-control or GFP-Nup358-MC1 as indicated. GFP-control or GFP-Nup358-MC1 was later induced by doxycycline, lysed and subjected to western blotting using indicated antibodies. The extent of Nup358 depletion and expression of GFP-proteins (left) and the relative levels of pAkt (right top) were monitored by western blotting using indicated antibodies. Right bottom: Quantitative representation depicting the relative levels of pAkt normalized to total Akt ($n = 3$ independent experiments). Data are mean ± SEM, Student's *t* test. *P* value is indicated. (F) Nup358-MC1 expression is sufficient to restrict ERMCSs. HeLa cells stably expressing inducible GFP-control or GFP-Nup358-MC1 were induced with doxycycline, fixed and subjected to PLA. Left: Representative images showing the extent of the interaction between VAPB and PTPIP51 as assessed by PLA (purple dots). Cell boundaries are demarcated with dotted lines. *Indicates low or no GFP-expressing cells. Scale bar, 10 μm. Right: Quantitative data depicting the number of PLA dots per cell GFP-positive cell under the indicated scenario. Number of Nup358–VAPB PLA dots per cell ($n = 90$ for both GFP-control and GFP-Nup358-MC1 from three independent experiments). Data are mean ± SD, Student's *t* test. *P* values are indicated. Source data are available online for this figure.

## Methods

### Mammalian cell culture and treatments

HeLa S3, HEK293T, U2OS and Huh7 cells were cultured in DMEM (10569010, Gibco/Invitrogen) with 10% FBS (16000044, Gibco/Invitrogen) and 10 μg/ml ciprofloxacin (Fresenius Kabi, India). The cell lines were routinely tested for mycoplasma contamination. Lipofectamine RNAimax (13778150, Invitrogen) was used for siRNA transfections, while polyethyleneimine, linear (PEI, MW-25,000; Polysciences Inc.) or Lipofectamine 2000 (11668019, Invitrogen) was used for transfection of plasmid constructs.

### Chemicals

The following reagents were used in the study: insulin (I6634 and I0516, Sigma), EGF (E9644, Sigma), Mifepristone-RU486 (M8046, Sigma), Wortmannin (W1628, Sigma), MitoTracker Green FM (M7514, Invitrogen), Puromycin (Invitrogen #A11138-03) and Polybrene (Sigma #107689).

### Antibodies

The following antibodies were used in this study. Anti-Nup358 (Joseph et al, 2004), (western blot (WB),1:3000; immunofluorescence (IF); proximity ligation assay (PLA), 1:600). Anti-mTOR (2983; WB, 1:4,000; PLA, 1:400), -Rictor (2140; PLA,1:100), Sin1 (12860; WB, 1:3000; PLA, 1:100), Mitofusin 2 (9482S; WB, 1:4000), -pAkt-S473 (9271; WB, 1:3000), -pAkt-T308 (9275; WB, 1:3000), -Akt (9272; WB, 1:3000), pGSK3β (5558; WB, 1:2000), pS6K (9205; WB, 1:3000), S6K (9202; WB, 1:3000), pS6 (2211; WB 1:4000), S6 (2317; WB, 1:4000) and GβL/mLST8 (3274; WB, 1:3000) were purchased from Cell Signalling Technology. Anti-Nup88 (6111896; WB, 1:3000), -Nup62 (610497; WB, 1:4000), -Tom20 (612278; IF, 1:200) and -GSK3β (610202; WB, 1:3000) were purchased from BD

biosciences. Anti-PDIA3 (AMAB90988; WB, 1:3000; IF, 1:500) was from Sigma and anti-RanGAP1 (33-0800; IF, 1:300; PLA, 1:500) was from Invitrogen. Anti-VAPB (66191-1-1 g; WB, 1:4,000; PLA, 1:900), -PTPIP51 (20641-1-AP; WB, 1:3000; PLA, 1:900), -α-tubulin (66031-1-Ig, WB, 1:5000), -GSK3β (22104-1-AP; WB, 1:2000; PLA, 1:1000), -BAP31 (11200-1-AP; PLA, 1:4000), -Fis1 (66635-1-Ig; PLA, 1:3000) and -Sin1 (15463-1-AP; WB, 1:3000) were purchased from Proteintech. Anti-Rictor (ab104838; WB, 1:2000), Nup214 (ab70497; IF, 1:100) were purchased from Abcam. Anti-myc (sc-401; WB, 1000), -Tom20 (sc17764; WB, 1:3000) and GAPDH (sc-166574; WB, 1:5000) were from Santa Cruz Biotechnology Inc. Other antibodies used in the studies were Anti-FLAG (F3165; WB, 1:5000) and -vinculin (V9131; WB, 1:10,000) from Sigma, anti-VAPB (VMA00454; PLA, 1:600) from Bio-Rad, Anti-IP3R3 (07-1213; PLA, 1:500) from Millipore, anti-HA (MMS-101R; WB, 1:5000) from Covance Research Products (BioLegend), anti-TNRC6A/GW182 (A302-329A; WB, 1:500) from Bethyl Laboratories and anti-MBP antibody from New England BioLabs (E8032S; WB, 1:3500). Anti-dNup358 (WB, 1:3000) and -PTPIP51 (PLA, 1:250) were produced in-house as detailed below.

### DNA constructs

pCI-neo-HA-PTPIP51 and pCI-neo-myc-VAPB were gifts from Christopher Miller (King's College London, UK). pCRISPR-CG01 (HCP216100-CG01-1-10) was purchased from GeneCopoeia and pcDNA3-Flag-mTOR wt (ID-26603) was procured from Addgene. Mito-BFP and RFP-KDEL (ER) were provided by Jennifer Lippincott-Schwartz (Janelia Research Campus, USA), pLVX-TetOne-Puro Vector (21915, Addgene), psPax2 (12260, Addgene), pMD2.G (12259, Addgene), Tet-pLKO-puro (Addgene #21915) and pcDNA3-hPACS2-3HA was from Gary Thomas (University of Pittsburgh, USA).

pEGFP-Nup358 (full length), pEGFP-siRES-Nup358, pEGFP-Nup358-N, pEGFP-Nup358-M and pEGFP-Nup358-C were described

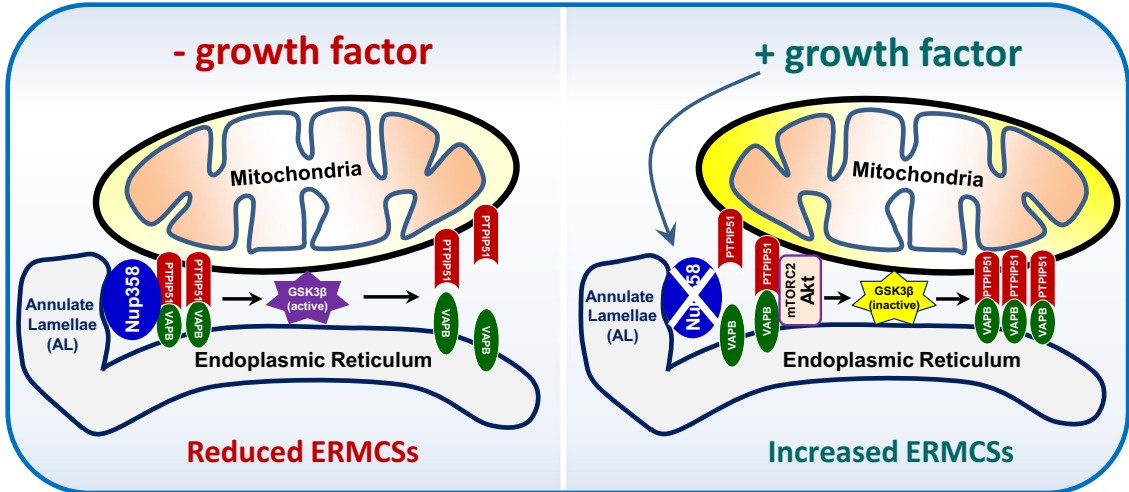

**Figure 7. Model for the role of Nup358 in ERMCS remodelling during growth factor signalling.**

Left: In the absence of growth factor signalling, presence of Nup358 at the ERMCSs due to its interaction with VAPB and PTPIP51 prevents the recruitment and activation of mTORC2 at the ERMCSs, leading to inhibition of Akt, activation of GSK3β and thereby decreased ER–mitochondria connectivity. Right: In the presence of growth factors, through an unknown (but PI3 kinase-independent) mechanism, interaction of Nup358 with VAPB/PTPIP51 is reduced. This allows the recruitment of mTORC2/Akt complex to the ERMCSs through interaction with preformed VAPB–PTPIP51 tether. This results in activation of mTORC2 and its downstream target Activated Akt which in turn phosphorylates and inhibits GSK3β. This prevents GSK3β-mediated disruption of VAPB–PTPIP51 interaction which subsequently increases ER–mitochondrial contacts. Thus, a reciprocal binding of Nup358 and mTORC2 with VAPB-PTPIP51 complex, ultimately determines the extent of ER–mitochondria connectivity by modulating the activity of GSK3β.

earlier (Joseph and Dasso, 2008; Sahoo et al, 2017). pEGFP-Nup358-MC1 (GFP-Nup358-MC1, amino acids 1949–2786) was generated from pEFGP-Nup358 full length by deleting sequences corresponding to both N-terminal and C-terminal regions using appropriate restriction digestion and self-ligation. The lentiviral construct expressing GFP-Nup358-MC1 was generated by subcloning the region encoding GFP-Nup358-MC1 from pEGFP-Nup358-MC1 into pLVX-TetOne-Puro.

Nup358-MC2 (2012–2771 aa) encoding region was PCR-amplified and cloned into BamHI/ XhoI sites of pGEX-6P1 vector (GE Healthcare). MBP-PTPIP51 (36–470 aa) was obtained by cloning the corresponding region of PTPIP51 into pMAL-c2 (New England BioLabs) at the appropriate sites to generate PTPIP51 containing MBP tag at N terminus and 6XHis tag at C terminus. MBP-VAPB (1–218 aa) was generated by PCR amplifying the corresponding region and cloning it into XmnI/XbaI sites of pMAL-c2 vector (GE Healthcare).

GSK3β was initially PCR-amplified from pcDNA3-hGSK3β-V5-WT (a kind gift from Thilo Hagen, National University of Singapore) and cloned into pEGFP-N1 vector (Clontech). For generation of pLVX-TetOne-GSK3β-GFP, GSK3β and GFP was subcloned from pEGFP-N1-GSK3β and pEGFP-C2, respectively, into pLVX-TetOne-Puro vector (21915, Addgene) using appropriate restriction digestions. The construct expresses GSK3β with GFP fused at its C terminus (GSK3β-GFP).

Constructs harbouring inducible shRNA control (shControl) and shRNA against Nup358 (shNup358) were generated by cloning the following target sequences into the Tet-pLKO-puro (Addgene #21915). shControl (5′-GTGGACTCTTGAAAGTACTAT-3′) and shNup358 (5′-GGTGAAGATGGATGGAATA-3′) were cloned into the Tet-pLKO-puro vector (Addgene #21915) at the AgeI/ EcoRI sites.

## siRNAs

The following sequences were used for siRNAs. siNup358 (5′-GG TGAAGATGGGATGGAATA-3′) (Sahoo et al, 2017), siControl 5′-T TCTCCGAACGTGTCACGT-3′) (Sahoo et al, 2017), siVAPB (5′-GCTCTTGGCTCTGGTGGTT-3′), siPTPIP51 (5′-CCTTAGACCT TGCTGAGAT-3′), siMfn2 (5′-GTGATGTGGCCCAACTCTA-3′), siRictor (5′-GCAGCCTTGAACTGTTTAA-3′) and ON-TARGET plus Human MAPKAP1/Sin1 (79109) siRNA SMARTpool (L-014315-00-0005) were purchased from Dharmacon. siRNAs against GW182 isoforms, siTNRC6A/GW182 (4390824, ID s26154) and siTNRC6B (4392420, IDs 23060), were obtained from Thermo-Fisher Scientific (Ambion). All siRNA transfections (40 nM, 72 h) were performed using Lipofectamine RNAiMAX (13778150, Invitrogen) as per the manufacturer's instruction.

## Co-immunoprecipitation (co-IP) and western blot (WB) analysis

In general, cell lysates for immunoblot analysis were prepared in 1% NP-40 buffer [20 mM Tris-HCl (pH 8), 137 mM NaCl, 10% glycerol, 2 mM EDTA, supplemented with 7.5 mM NaF, 0.75 mM sodium orthovanadate, 1 mM PMSF, protease inhibitor cocktail of leupeptin (50 μg/ml), aprotinin (5 μg/ml), and pepstatin (2 μg/ml)]. After thorough retro pipetting, the cell lysates were centrifuged at 12,900× g for 10 min to pellet the cell debris. Clear supernatant was collected and protein concentration was estimated using Bradford (Bio-Rad) assay. Samples were subjected to SDS-PAGE and transferred to PVDF membrane (Millipore) for western analysis.

For extracting phosphoproteins, cell lysis was performed in Phospho-lysis buffer [50 mM Tris-HCl (pH 7.5), 1 mM EDTA, 1%

Triton X-100, 10% glycerol] or RIPA buffer [50 mM Tris-HCl (pH 7.4), 150 mM NaCl, 1% NP-40, 0.1% SDS, 0.5% sodium deoxycholate], supplemented with 1 mM DTT, 50 mM NaF, 5 mM sodium pyrophosphate, 1 mM PMSF and protease inhibitor cocktail (Roche).

For immunoprecipitation of endogenous PTPIP51, Nup358 or ectopically expressed GFP-tagged proteins, the cell extracts were made in 1% NP-40 cell lysis buffer as previously described. The whole cell extracts were incubated with Protein A sepharose beads (101041, Invitrogen) coated with anti-PTPIP51, anti-Nup358 or anti-GFP antibody for 2 h at 4 °C. The precipitates were washed once with 1% NP-40 cell lysis buffer, followed by two washes with tris-buffered saline (TBS). The IP samples were subjected to SDS-PAGE and analysed by WB.

For IP of mTOR, cells were lysed in 0.5% CHAPS buffer containing 120 mM NaCl, 40 mM HEPES (pH 7.5), 0.2 mM EDTA, 10% glycerol, and FLAG-mTOR was immunoprecipitated with anti-FLAG beads (EZview™ Red ANTI-FLAG® M2 Affinity Gel, Sigma, F3165). The immunoprecipitates were washed three times with 0.5% CHAPS buffer before proceeding with WB.

## Subcellular fractionation—ERMCS isolation

ERMCS (MAM) fractionation was performed from HeLaS3 cells based on the standardized MAM isolation protocol (Williamson et al, 2015; Wieckowski et al, 2009). Cells from six T-175 confluent flasks were trypsinized and centrifuged at 600 *g* for 5 min. All the further steps were performed at 4 °C using fractionation buffers supplemented with a protease inhibitor cocktail. The cell pellet was resuspended in IB-1 buffer [225 mM mannitol, 75 mM sucrose, 0.1 mM EGTA and 30 mM Tris-HCl (pH 7.4)] and homogenized by passing through syringes of different nozzle sizes (21Ga, 23Ga and 24Ga), 10 times each. Post nuclear fraction (PNF) was collected from the homogenate by centrifugation at 600× *g* for 5 min. Later, crude mitochondria was pelleted from the PNF by centrifuging at 7000× *g* for 10 min and the supernatant containing soluble proteins and ER/microsomes was collected as cytoplasmic fraction. To avoid the microsomal contamination, the crude mitochondrial pellet was resuspended in IB-2 buffer [225 mM mannitol, 75 mM sucrose, and 30 mM Tris-HCl (pH 7.4)] and centrifuged at 7000× *g* for 10 min. This step was repeated once again, centrifuging at 10,000× *g*, this turn. The final pellets were washed and reconstituted in MRB buffer [250 mM mannitol, 5 mM HEPES (pH 7.4), and 0.5 mM EGTA)] and gently layered on top of the percoll medium [225 mM mannitol, 25 mM HEPES (pH 7.4), 1 mM EGTA, and 30% percoll]. After 30 min of centrifugation at 95,000× *g*, the ERMCS (MAM) fraction, a whitish diffused band at the middle of tube, was carefully collected with a Pasteur pipette. The remaining lower layer, containing mitochondrial suspension, was diluted in MRB solution, and mitochondria were extracted by centrifugation at 6300× *g* for 20 min. Similarly, the ERMCS fraction was resuspended in MRB buffer and centrifuged at 100,000× *g* for 60 min to obtain the ERMCS pellet. All final fractions were dissolved in RIPA buffer [50 mM Tris-HCl (pH 7.4), 150 mM NaCl, 1% NP-40, 0.1% SDS and 0.5% sodium deoxycholate] and analysed by western blotting.

## Immunofluorescence (IF) and proximity ligation assay (PLA)

Cells seeded on coverslips were fixed using methanol (−20 °C) for 5 min. After washing with phosphate-buffered saline (PBS), cells were incubated in primary antibodies diluted in 2% normal horse serum (NHS, Vector Laboratories) for 60 min. After a PBS wash, Alexa Fluor labelled secondary antibodies (Molecular probes, ThermoFisher Scientific) diluted in 2% NHS were added and incubated for 30 min. For staining of the nucleus, Hoechst 33342 dye (Sigma) was added to the secondary antibody solution. After three washes with PBS, the coverslips were mounted with Vectashield medium (Vector Laboratories). Images were acquired using Olympus FV3000 confocal laser scanning microscope with the ×60 (1.42 NA) or ×100 (1.45 NA) objective. The images were processed with cellSens (ver.2.3) constrained iterative deconvolution for better resolution of ER and mitochondrial network.

Proximity Ligation assay (PLA) was used to identify in situ protein interactions. Briefly, cells grown on coverslips were fixed with methanol (−20 °C) for 5 min and probed with the respective primary antibodies. PLA was performed using Duolink reagents (Sigma; DUO92002; DUO92004) as described in Duolink® PLA Fluorescence Protocol (Sigma). The PLA signals developed by Duolink® In Situ Detection Reagents Red (DUO92101) or Duolink® In Situ Detection Reagents Green (DUO92014) were detected and acquired by Olympus FV 3000 microscope and quantitated using NIH ImageJ or CellProfiler (Irvine et al, 2004). Phase-contrast images were used to demarcate the cell boundaries.

## Stimulated emission depletion microscopy (STED)

Huh7 cells grown on high-precision microscope cover glasses (Marienfeld, Germany) were fixed with 4% paraformaldehyde (20 min), followed by permeabilization with 0.1% Triton X-100 (10 min). The cells were then incubated with primary antibodies diluted in 2% NHS. After washing the coverslips thrice with PBS, secondary antibodies (Alexa Flour 568 and Alexa Flour 657) prepared in 2% NHS were added, washed thrice with PBS and the coverslips were mounted in Prolong Gold (Invitrogen). Images were acquired using Leica TCS SP8 STED 3× with the ×100, 1.4 NA objective.

## Quantitation of ER and mitochondria colocalization

RFP-ER and Mito-BFP plasmids were co-expressed to visualize the contacts between ER and mitochondria in live cells. Images were analysed for co-localisation between ER and mitochondria using ImageJ JACoP software. Manders' coefficient was calculated from the fraction of mitochondria overlapping with the ER. To visualize the overlapping pixels, represented as ERMCSs, ImageJ Co-localisation Finder was used.

## Transmission electron microscopy

For ultrastructure analysis, the monolayer culture cells were washed with 1× PBS and fixed with 3% glutaraldehyde for 2 h at 40 °C and post fixed in 1% aqueous osmium tetroxide for 1 h at 4 °C, followed by en block staining in 2% aqueous uranyl acetate. The cells were then subsequently dehydrated in increasing grade of alcohol for 15 min each. The cells were further infiltrated with an increasing series of hydroxypropyl methacrylate (HPMA) followed by Epon resin infiltration for 15 min each and polymerised in Epon resin.

The Epon sheet was then cut into small pieces and re-embedded on the pre- polymerized block. Ultrathin sections of 70 nm were cut and contrasted with lead citrate and images were acquired using JEM 1400 Plus Transmission Electron Microscope (JEOL, Japan) at 120 kV.

For quantitation of ERMCS from the electron micrographs, regions of mitochondria at a distance of ≤30 nm from the ER were identified using ImageJ (Lam et al, 2021). The percentage of mitochondrial perimeter associated with ER was then determined using ImageJ.

## Generation of Nup358 knockout (KO) cell lines

HeLa Nup358 KO cells were generated using CRISPR-Cas9 technology. HeLa cells were co-transfected with Nup358 CRISPR-Cas9 (pCRISPR-CG01, GeneCopoeia) and pTSiN-puro-Cre constructs. Transfected cells were selected for puromycin resistance. Single-cell clones were obtained, which were verified for Nup358 KO by IF and immunoblotting. The sgRNA target region within the *Nup358* gene was PCR-amplified and sequenced to confirm that the genomic manipulation occurred (Fig. EV4).

## Generation of inducible HEK293T and HeLa lines expressing control and Nup358-specific shRNA

Stable inducible shRNA HEK293T cell lines for shControl and shNup358 were generated using lentivirus transduction. To generate lentivirus, HEK293T cells were co-transfected with psPAX2 (Addgene #12260) and pMD2.G (Addgene #12259) along with the shRNA-specific Tet-pLKO-puro construct, using Lipofectamine 2000 (Invitrogen). The cells were replenished with fresh medium the following day, and the conditioned medium containing lentivirus was collected after 48. HEK293T and HeLa cells were transduced with the lentivirus-containing medium diluted 1:1 with DMEM and a final concentration of 8 µg/ml Polybrene. Forty-eight hours post transduction, the stable cell pool, resistant to puromycin (2.5 µg/ml), was selected.

## Generation of inducible HeLa lines expressing GFP-control, GFP-Nup358-MC1 and GSK3β-GFP

The pLVX-TetOne-Puro Vector (21915, Addgene) containing a doxycycline-inducible CMV promoter and puromycin resistance gene was utilized to generate stable GFP-control, GFP-Nup358-MC1 or GSK3β-GFP-expressing HeLa S3 cells. For this, HEK293T cells were seeded at a density of 700,000 cells in a 35-mm cell culture plate. After cell adhesion, individual transfections were conducted using 1.9 µg of pLVX-GFP, pLVX-GFP-Nup358-MC1 or pLVX-GSK3β-GFP plasmid, along with 2.14 µg of PsPax2 and 0.56 µg of PMD2.G plasmids, using Lipofectamine 2000. Twenty-four hours post transfection, the medium was replaced with fresh medium, and the viral soup was collected after 24 h. The collected viral soup was filtered through a 0.45-µm filter and mixed with 10% FBS-containing DMEM in a 1:1 ratio. This mixture was added to HeLa S3 cells, which were seeded a day before, along with 8 µg/ml of polybrene. The medium containing the viral soup and polybrene was replaced again in the next 24 h and subsequently in the subsequent 24 h. After the final transduction, the medium was replaced with 10% FBS-containing DMEM supplemented with

puromycin. The cells were then subjected to puromycin selection for a week to obtain stable cell lines expressing the desired proteins. Confirmation of the stable cell line expressing the desired proteins was done through doxycycline induction and western blotting.

## Treatments

In general, prior to western analysis, cells were replenished with fresh medium containing 10% FBS for 24 h. In case when growth factor (insulin and EGF) signalling was assessed under siRNA-treated conditions, cells were initially transfected with siRNA for 69 h and the old medium was replaced with serum-free DMEM for 3 h (for western analysis) or 12 h (for PLA). In the case of rescue experiments using stable inducible HEK293T shControl and shNup358 cells, pEGFP-MBP-control or pEGFP-Nup358-siRES was transfected for 24 h following which the cells were kept in fresh DMEM containing 1% FBS for 24 h before processing for WB. For WB and PLA analyses of inducible GFP-control or GFP-Nup358-MC1 HeLa cells, the cells were seeded in and cultured for 36 h in DMEM containing 10% FBS and doxycycline (1 µg/ml). For PI3 kinase inhibition experiments, cells were cultured in DMEM containing 10% FBS for 36 h post which the medium was replaced with serum-free DMEM for 9 h. The cells were then treated with 2.5 µM wortmannin for 3 h post which the medium was replaced with serum-free medium with/without 2 nM insulin.

## Antibody production

Antibody against human PTPIP51 was raised in rabbits. Bacterially expressed and purified His-tagged PTPIP51 fragment corresponding to amino acids 244–470 was used as the antigen. Specific antibodies were affinity purified using the antigen and used in this study. Antibodies for Drosophila Nup358 were generated in mice against a region corresponding to amino acids 2319–2555 (variant B), expressed and purified from bacteria.

## Purification of recombinant proteins

Nup358-MC2 protein was expressed in E.coli BL21 CodonPlus RIL (DE3) cells and the culture was induced with 0.3 mM isopropyl-ᴅ-thiogalactoside at 18 °C overnight. The cells were lysed in 50 mM Tris-HCl (pH 8.0), 150 mM NaCl, 0.1% NP-40, 1 mM dithiothreitol, 1 mM EDTA,1 mM phenylmethylsulfonyl fluoride and 20 mg/ml concentrated lysozyme (2.5 µl/ml). Sonication with 70% amplitude, 2 s pulse on −15 s pulse off for 1 min was used to lyse the cells. After centrifugation at 12,900× *g*, 4 °C for 30 min, the supernatant was added to Glutathione Agarose Beads (Bio Bharati Life Sciences). The protein was eluted with 50 mM Tris-HCl (pH 8.0), 30 mM reduced glutathione and 1 mM PMSF.

PTPIP51 (36–470 aa) was expressed in *E. coli* BL21 Star (DE3) cells and induced with 0.5 mM isopropyl-ᴅ-thiogalactoside at 18 °C for ~5 h. Cells were resuspended in lysis buffer containing 50 mM Tris-HCl (pH 8.0), 300 mM NaCl, 0.1% NP-40, 5% glycerol, 1 mM dithiothreitol, 1 mM phenylmethylsulfonyl fluoride and 20 mg/ml concentrated lysozyme (2.5 µl/ml) and sonicated at 70% amplitude, 2 s pulse on −15 s pulse off for 1 min. The supernatant was first added to Ni-NTA resin (Invitrogen), and the protein was eluted

using 250 mM imidazole in the lysis buffer. The eluted protein was further bound to Amylose resin (New England BioLabs), followed by washing. The final elution of the protein was done using 15 mM maltose in the lysis buffer.

VAPB (1–218 aa) was expressed *E. coli* BL21 CodonPlus RIL (DE3) cells and induced with 0.5 mM isopropyl-D-thiogalactoside at 37 °C for ~6 h. Cells were lysed in 50 mM Tris-HCl (pH 8.0), 300 mM NaCl, 0.1% NP-40, 5% glycerol, 1 mM dithiothreitol, 1 mM phenylmethylsulfonyl fluoride and 20 mg/ml concentrated lysozyme (2.5 µl/ml) and sonicated at 70% amplitude, 2 s pulse on −15 s pulse off for 1 min. The supernatant was added to the Amylose resin (New England BioLabs), and the protein elution was done using 15 mM maltose in lysis buffer.

All recombinant proteins were dialysed in 1× TBS, 5% glycerol and 1 mM PMSF.

### In vitro protein–protein interaction assay

Magnetic Protein G Dynabeads (10004D, ThermoFisher Scientific) were bound with 2 µg of monoclonal anti-MBP antibody (E8032S, New England BioLabs). Meanwhile, 2 µM of MBP (Control), MBP-PTPIP51 or MBP-VAPB was incubated with 1.5 µM of GST-Nup358-MC2 in the assay buffer (in a reaction volume of 100 µl) containing 25 mM Tris-HCl (pH 7.4), 150 mM NaCl, 5% glycerol, 0.01% v/v Tween20 for 2 h at 4 °C with gentle retro pipetting. Later, the MBP-tagged recombinant proteins were immunoprecipitated and the beads were washed with the assay buffer, and the IP samples were subjected to western blotting.

### MitoTracker Green FM staining

HeLa cells transfected with siControl or siNup358 were trypsinized and washed with PBS. One million cells were resuspended in 0.5 ml of PBS and 50 nM MitoTracker Green FM (M7514, Invitrogen) was added, followed by gentle pipetting for mixing. The Cells were then incubated at 37 °C in $CO_2$ incubator for 15 min. Cells were washed again with PBS to remove any unbound dye. Unstained cells were used as control. FACS analysis was performed using BD FACSCanto™ II System.

### Drosophila strains and rearing

Flies were maintained on standard Cornmeal Agar at 25 °C and 65% relative humidity in a 12 h light: dark cycle. Gal4/UAS system was used to induce the expression of our gene of interest. Fly strains used in our experiments were Cg-Gal4 (BDSC: 7011), Elav-Gal4 (BDSC: 458), Elav-Gene Switch (BDSC: 43642), UAS-Nup358 RNAi (BDSC: 34967), UAS-Luc Val10 RNAi (BDSC: 35788).

### Drosophila gene switch expression

F1 progeny from the cross of Elav-Gene Switch with either UAS-Nup358 RNAi or UAS-Luc Val10 RNAi were collected upon eclosion. The progeny were starved overnight in vials containing moist paper and fed 500 µM mifepristone (RU486) dissolved in 80% ethanol for 72 h post starvation. Later, the fly heads ($n = 5$) were chopped and lysed in S2 buffer [100 mM NaCl, 50 mM Tris-HCl (pH 7.5), 0.1% NP-40 with protease inhibitors] and processed for western blotting.

### Statistics and reproducibility

All statistical analyses were done in GraphPad Prism version 8, calculated from independently repeated experiments as indicated in the respective Figure Legends. For western blot analysis, values were considered from at least three independent experiments ($n = 3$) and a two-sided unpaired Student's $t$ test was performed. The values were graphically expressed as mean ± SEM and $P$ values were as indicated in legends. All immunostaining data was also analysed by a two-sided unpaired Student's $t$ test. The mean ± SD was schematically represented and the level of significance is as stated in the Figure Legends. In all plots, $P$ values are as indicated; $P \leq 0.05$ are considered significant and $P > 0.5$, not significant.

## Data availability

This study includes no data deposited in external repositories.

The source data of this paper are collected in the following database record: biostudies:S-SCDT-10_1038-S44319-024-00204-8.

## Peer review information

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

## Acknowledgements

The authors thank C. Miller, T. Hagen and J. Lippincott-Schwartz for sharing reagents; NCCS Bio-Imaging facility staff for help with microscopy; Richa Ricky (IISER, Pune) for critical comments on the manuscript; and members of Joseph and Seshadri Lab for insightful discussions. Stocks obtained from the Bloomington Drosophila Stock Centre (NIH P40OD018537) were used in this study. We thank Girish Ratnaparkhi, IISER, Pune, for sharing UAS-Nup358 RNAi line and Swagatika Panigrahi for the help in generating dNup358 antibody and generating a few constructs. Electron Microscope facility, ACTREC, Tata Memorial Centre, Kharghar, Navi Mumbai is acknowledged for the help with transmission electron microscopy. This work was supported by grants from Department of Biotechnology, Ministry of Science and Technology, India (BT/PR27451/BRB/10/1655/2018), Department of Science and Technology, Ministry of Science and Technology, Government of India (SPR/2021/000352) and Pratiksha Trust Extra-Mural Support for Transformational Aging Brain Research (EMSTAR) (EMSTAR/20230-03) and intramural funding from National Centre for Cell Science to JJ. Financial support from the Council of Scientific and Industrial Research, Ministry of Science and Technology, Government of India to MK and LO; from Department of Biotechnology, Ministry of Science and Technology, India to RS, YB. and PV and from University Grants Commission (UGC), Government of India to AL through research fellowships, is gratefully acknowledged. The authors dedicate this article to the former Director of NCCS, Late Dr. Mohan Wani, for all the encouragement and his immense support towards the publication of this manuscript.

## Author contributions

**Misha Kalarikkal**: Conceptualization; Resources; Data curation; Formal analysis; Validation; Investigation; Visualization; Methodology; Writing—original draft; Writing—review and editing. **Rimpi Saikia**: Conceptualization; Resources; Data curation; Formal analysis; Validation; Investigation; Visualization; Methodology; Writing—original draft; Writing—review and editing. **Lizanne Oliveira**: Conceptualization; Resources; Data curation; Formal analysis; Validation; Investigation; Visualization; Methodology; Writing—original draft; Writing—review and editing. **Yashashree Bhorkar**: Resources; Data curation; Formal analysis; Validation; Investigation; Visualization; Methodology; Writing—review and editing. **Akshay Lonare**: Resources; Data curation; Formal analysis; Validation; Investigation; Visualization; Methodology; Writing—review and editing. **Pallavi Varshney**: Conceptualization; Data curation; Formal analysis; Validation; Methodology; Writing—review and editing. **Prathamesh Dhamale**: Resources; Data curation; Investigation; Methodology; Writing—review and editing. **Amitabha Majumdar**: Resources; Data curation; Formal analysis; Validation; Investigation; Methodology; Writing—review and editing. **Jomon Joseph**: Conceptualization; Resources; Data curation; Formal analysis; Supervision; Funding acquisition; Validation; Investigation; Visualization; Methodology; Writing—original draft; Project administration; Writing—review and editing.

Source data underlying figure panels in this paper may have individual authorship assigned. Where available, figure panel/source data authorship is listed in the following database record: biostudies:S-SCDT-10_1038-S44319-024-00204-8.

## Disclosure and competing interests statement

The authors declare no competing interests.

# Expanded View Figures

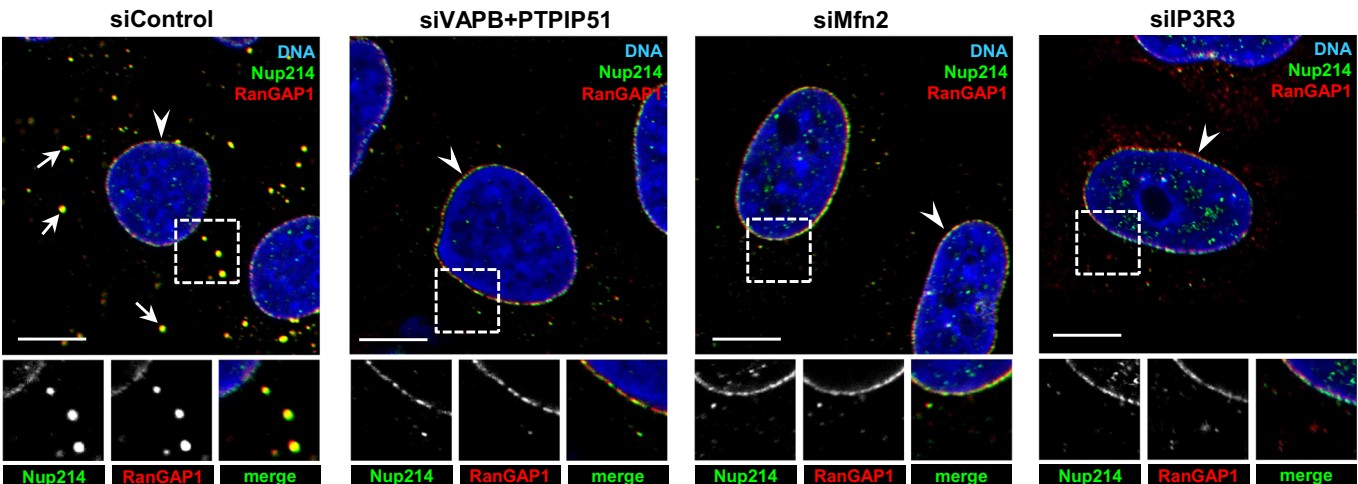

**Figure EV1. ERMCS integrity is important for AL assembly and/or stability.**

Depletion of ERMCS proteins affects AL assembly. HeLa cells were treated with indicated siRNAs and the AL integrity was monitored by presence of cytoplasmic Nup214 (green) and RanGAP1 (red). Here RanGAP1 was used as a surrogate for Nup358, as RanGAP1 is known as a strong binding partner that colocalizes with Nup358 at the nuclear envelope and AL. Number of cytoplasmic puncta, which represent AL (shown in arrows), are significantly reduced when ERMCS proteins are depleted as compared to control cells. Under the same conditions, the NE staining of nucleoporins (arrowheads) was largely unaffected. Scale bar, 10 μm.

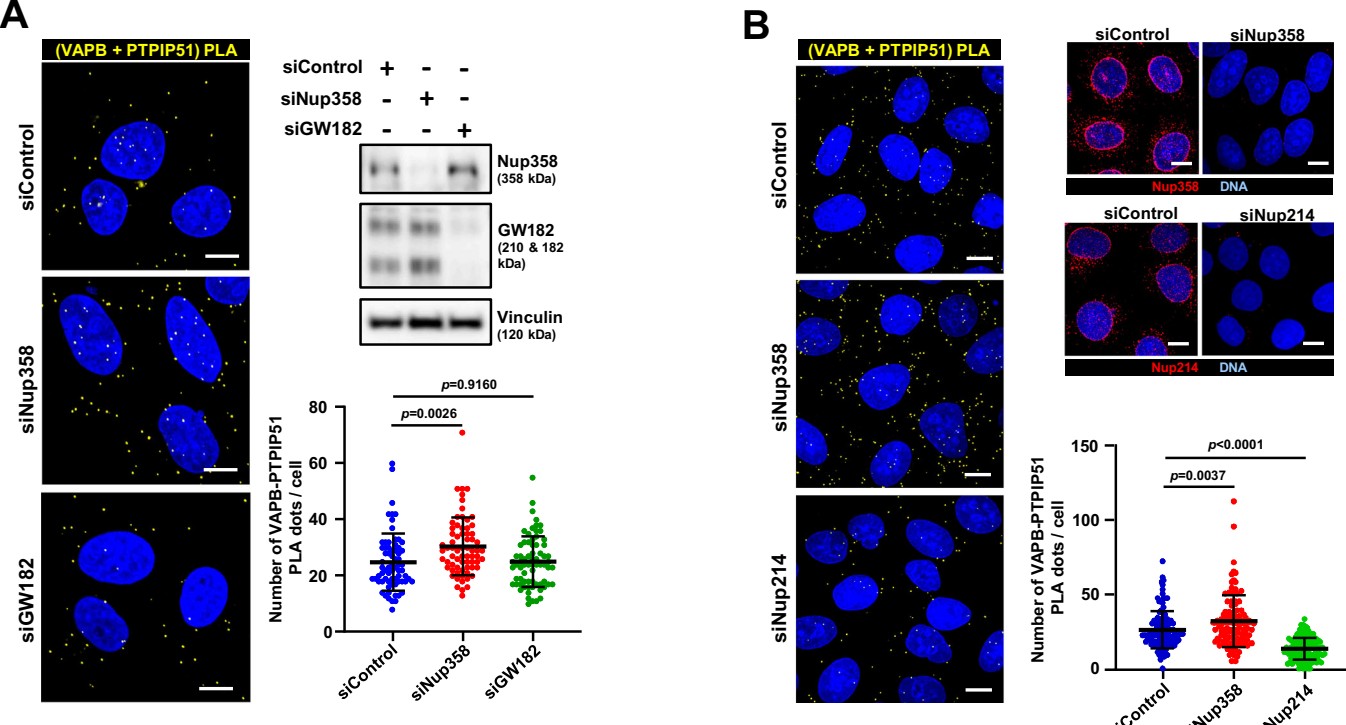

**Figure EV2. Interfering with miRNA pathway does not affect ERMCSs.**

(A) Depletion of GW182 does not affect contacts between ER and mitochondria. HeLa cells were treated with indicated siRNAs and the intactness of ERMCS was monitored in situ by PLA using VAPB and PTPIP51 antibodies. Left: Representative microscopic images showing PLA puncta (yellow) under the indicated conditions. DNA was stained with Hoechst 33342 (blue). Scale bar, 10 μm. Right top: Depletion of indicated proteins was confirmed by western blotting. Vinculin was used as loading control. Right bottom: Quantitative data showing number of PLA dots per cell, derived from indicated conditions ($n = 70$ cells for siControl; 69 cells for siNup358 and 71 cells for siGW182 from 3 independent experiments). Data are mean ± SD, unpaired Student's $t$ test. $P$ values are indicated. (B) Depletion of Nup214 reduces contacts between ER and mitochondria. HeLa cells were treated with indicated siRNAs and the ERMCS were monitored in situ by PLA using VAPB and PTPIP51 antibodies. Left: Representative microscopic images showing PLA puncta (yellow) under the indicated conditions. DNA was stained with Hoechst 33342 (blue). Scale bar, 10 μm. Right top: Depletion of indicated proteins was confirmed by immunostaining with the indicated antibodies (red). Right bottom: Quantitative data showing number of PLA dots per cell, derived from indicated conditions ($n = 127$ cells for siControl; 106 cells for siNup358 and 116 cells for siNup214 from 3 independent experiments). Data are mean ± SD, unpaired Student's $t$ test. $P$ values are indicated.

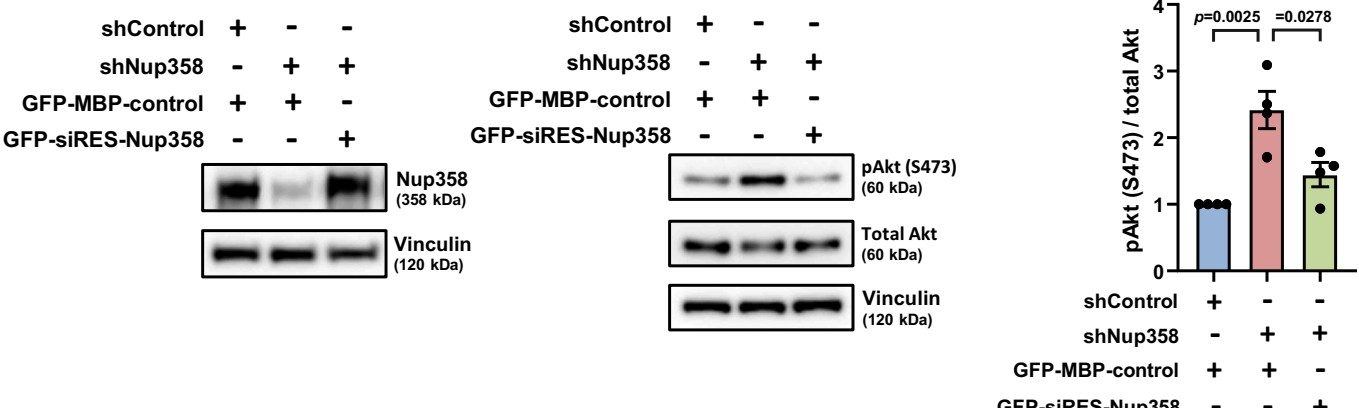

**Figure EV3. Ectopic expression of Nup358 rescues the increased mTORC2/Akt activity in Nup358-deficient cells.**

Stable HEK293T cells with Inducible short hairpin RNA (shRNA) for control (shControl) or Nup358 (shNp358) were initially induced with doxycycline. Later the cells were transfected with GFP-MBP-control or GFP-siRNA-resistant (siRES)-Nup358 construct as indicated. Left: Expression levels of Nup358 in the indicated samples were monitored by western blotting with Nup358 antibodies. Vinculin was used as loading control. Middle: Levels of indicated proteins were assessed by western blotting. Vinculin was loading control. Right: Quantitative representation depicting relative levels of pAkt normalized to total Akt ($n = 4$ independent experiments) under the indicated conditions. Data are mean ± SEM, Student's *t* test. *P* values are indicated.

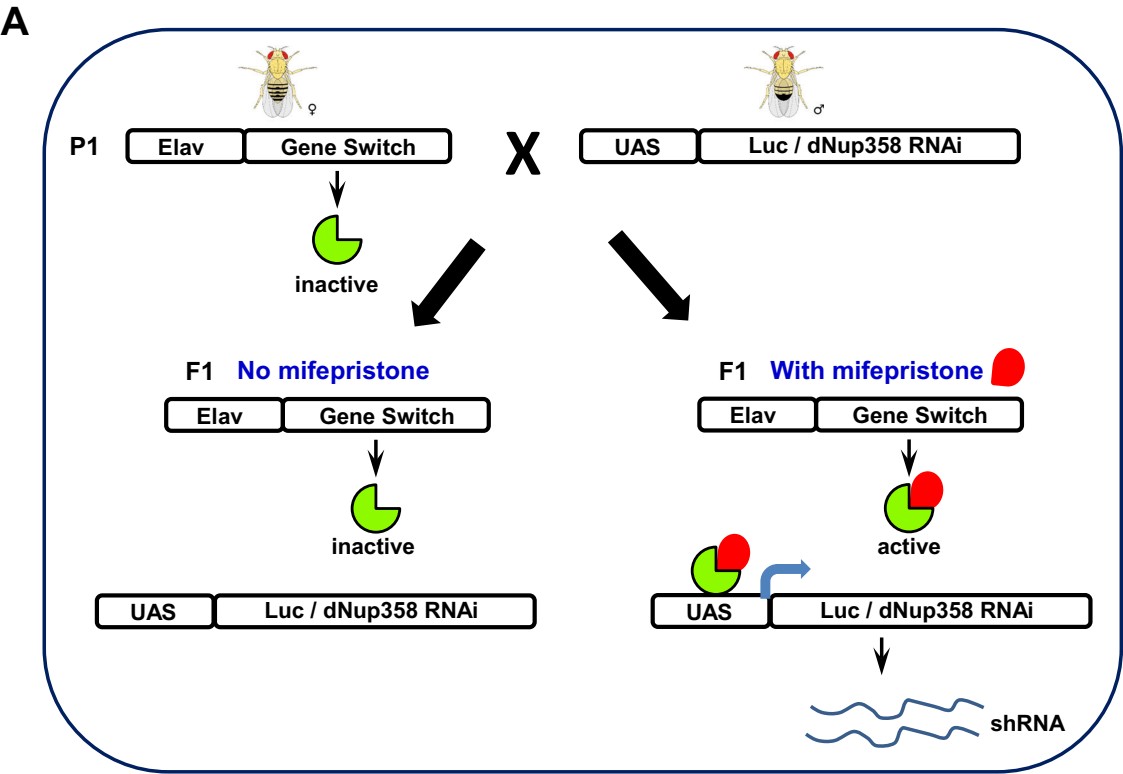

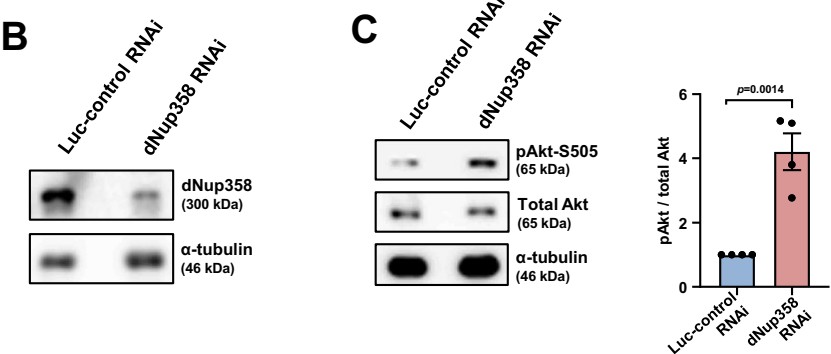

**Figure EV4.  Conserved regulatory role of Nup358 in restricting mTORC2/Akt activation in Drosophila.**

(**A**) A schematics showing how F1 progeny was obtained by crossing parental (P) lines as indicated. The adult Drosophila flies (F1) were treated with 500 µM mifepristone (RU486) for 72 h to induce control shRNA (Luciferase, Luc) or dNup358-specific shRNA. (**B**) Brains lysates were assessed for dNup358 knockdown by western blotting. α-tubulin was used as loading control. (**C**) Left: Lysates were analysed for pAkt (S505) levels by western blotting. α-tubulin was used as loading control. Right: Quantitative data showing the levels of pAkt (normalized to total Akt) under indicated conditions ($n = 4$ independent experiments). Data are mean ± SEM, unpaired Student's $t$ test. $P$ value is indicated.

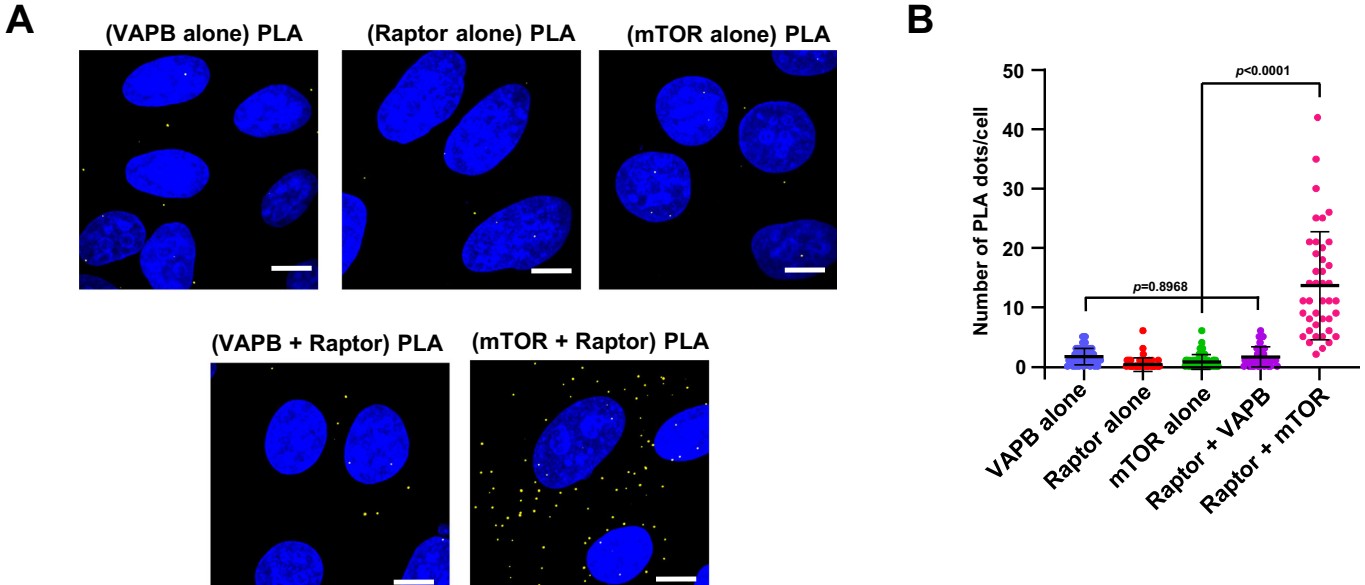

**Figure EV5. VAPB does not interact with the mTORC1-specific subunit Raptor.**

(A) HeLa cells were subjected to PLA (yellow dots) for detecting the VAPB-Raptor interaction and mTOR-Raptor (positive control) interaction, with individual antibody as negative controls. DNA was stained with Hoechst 33342 (blue). Scale bar, 10 μm. (B) Quantitation of PLA dots per cell ($n = 56$ cells for VAPB alone; 41 cells for Raptor alone; 57 cells for mTOR alone; 48 cells for VAPB-Raptor interaction; 41 cells for VAPB–mTOR interaction; from a single experiment). Data are mean ± SD, unpaired Student's *t* test. *P* values are indicated.

