## [Peer Review File · EMBO Reports]

Nup358 restricts ER-mitochondria connectivity by modulating mTORC2/Akt/GSK3 β signalling

Misha Kalarikkal, Rimpi Saikia, Lizanne Oliveira, Yashashree Bhorkar, Akshay Lonare, Pallavi Varshney, Prathamesh Dhamale, Amitabha Majumdar, and Jomon Joseph

Corresponding author(s): Jomon Joseph (josephj@nccs.res.in)

Review Timeline:

Submission Date:	30th Aug 23
Editorial Decision:	6th Dec 23
Revision Received:	7th May 24
Editorial Decision:	17th Jun 24
Revision Received:	24th Jun 24
Accepted:	27th Jun 24

Transaction Report:

Dear Joe,

Thank you for the submission of your research manuscript to our journal. I apologize for the unusual delay in handling your manuscript, but we have meanwhile received the full set of referee reports that is copied below.

As you will see, the referees came to different conclusions regarding the advance provided and the strength of the data and conclusions made. Referee 1 is an expert in mTORC signaling and considers the data interesting and overall convincing but also raises a few concerns that should be addressed. Referee 2 raises an important concern that is also shared by the other referees: i.e., the limited evidence for the involvement of annulate lamellae that is mainly based on Nup358 alone. This concern needs to be addressed with either more data or the statement needs to be toned down and focussed on Nup358. Please also address the concerns from referee 4 regarding the evidence for ERM stability and number.

From these comments it is clear that a major revision with significant work will be required before the manuscript can be considered for publication in EMBO Reports. It will be necessary to provide further insight into the connection to mTORC2, to confirm the effect on ERM stability and the number of contacts, the interaction with PTPN12 and VAPB, and to clarify the involvement of AL, as outlined above. That said and based on the support from three of the four referees, I would like to give you the opportunity to address the concerns and submit a revised version of your manuscript with the understanding that the referee concerns (as detailed above and in their reports) must be fully addressed and their suggestions taken on board. Please address all referee concerns in a complete point-by-point response. Acceptance of the manuscript will depend on a positive outcome of a second round of review. It is EMBO Reports policy to allow a single round of revision only and acceptance or rejection of the manuscript will therefore depend on the completeness of your responses included in the next, final version of the manuscript.

We realize that it is difficult to revise to a specific deadline. In the interest of protecting the conceptual advance provided by the work, we recommend a revision within 3 months (March 6th). Please discuss the revision progress ahead of this time with the editor if you require more time to complete the revisions.

I am also happy to discuss the revision further via e-mail or a video call, if you wish.

*******IMPORTANT NOTE:**

We perform an initial quality control of all revised manuscripts before re-review. Your manuscript will FAIL this control and the handling will be delayed IN CASE the following APPLIES:

- 1) A data availability section providing access to data deposited in public databases is missing. If you have not deposited any data, please add a sentence to the data availability section that explains that.
- 2) Your manuscript contains statistics and error bars based on $n=2$. Please use scatter blots in these cases. No statistics should be calculated if $n=2$.

When submitting your revised manuscript, please carefully review the instructions that follow below. Failure to include requested items will delay the evaluation of your revision. *****

- 1) a .docx formatted version of the manuscript text (including legends for main figures, EV figures and tables). Please make sure that the changes are highlighted to be clearly visible.
- 2) individual production quality figure files as .eps, .tif, .jpg (one file per figure). Please download our Figure Preparation Guidelines (figure preparation pdf) from our Author Guidelines pages <https://www.embopress.org/page/journal/14693178/authorguide> for more info on how to prepare your figures.
- 3) a .docx formatted letter INCLUDING the reviewers' reports and your detailed point-by-point responses to their comments. As part of the EMBO Press transparent editorial process, the point-by-point response is part of the Review Process File (RPF), which will be published alongside your paper.

- 4) a complete author checklist, which you can download from our author guidelines (<https://www.embopress.org/page/journal/14693178/authorguide>). Please insert information in the checklist that is also reflected in the manuscript. The completed author checklist will also be part of the RPF.
- 5) Please note that all corresponding authors are required to supply an ORCID ID for their name upon submission of a revised manuscript (<https://orcid.org/>). Please find instructions on how to link your ORCID ID to your account in our manuscript tracking system in our Author guidelines (<https://www.embopress.org/page/journal/14693178/authorguide#authorshipguidelines>)
- 6) We replaced Supplementary Information with Expanded View (EV) Figures and Tables that are collapsible/expandable online. A maximum of 5 EV Figures can be typeset. EV Figures should be cited as 'Figure EV1, Figure EV2' etc... in the text and their respective legends should be included in the main text after the legends of regular figures.
- For the figures that you do NOT wish to display as Expanded View figures, they should be bundled together with their legends in a single PDF file called *Appendix*, which should start with a short Table of Content. Appendix figures should be referred to in the main text as: "Appendix Figure S1, Appendix Figure S2" etc. See detailed instructions regarding expanded view here: <https://www.embopress.org/page/journal/14693178/authorguide#expandedview>
 - Additional Tables/Datasets should be labeled and referred to as Table EV1, Dataset EV1, etc. Legends have to be provided in a separate tab in case of .xls files. Alternatively, the legend can be supplied as a separate text file (README) and zipped together with the Table/Dataset file.
- 7) Please note that a Data Availability section at the end of Materials and Methods is now mandatory. In case you have no data that requires deposition in a public database, please state so instead of refereeing to the database. See also <https://www.embopress.org/page/journal/14693178/authorguide#dataavailability>). Please note that the Data Availability Section is restricted to new primary data that are part of this study.
- 8) At EMBO Press we ask authors to provide source data for the main figures. Our source data coordinator will contact you to discuss which figure panels we would need source data for and will also provide you with helpful tips on how to upload and organize the files.

Additional information on source data and instruction on how to label the files are available <https://www.embopress.org/page/journal/14693178/authorguide#sourcedata>.

10) Figure legends and data quantification:
The following points must be specified in each figure legend:

- the name of the statistical test used to generate error bars and P values,
 - the number (n) of independent experiments (please specify technical or biological replicates) underlying each data point,
 - the nature of the bars and error bars (s.d., s.e.m.)
- If the data are obtained from n {less than or equal to} 5, show the individual data points in addition to the SD or SEM.
 - If the data are obtained from n {less than or equal to} 2, use scatter blots showing the individual data points.

See also the guidelines for figure legend preparation: <https://www.embopress.org/page/journal/14693178/authorguide#figureformat>

11) Our journal encourages inclusion of *data citations in the reference list* to directly cite datasets that were re-used and obtained from public databases. Data citations in the article text are distinct from normal bibliographical citations and should directly link to the database records from which the data can be accessed. In the main text, data citations are formatted as follows: "Data ref: Smith et al, 2001" or "Data ref: NCBI Sequence Read Archive PRJNA342805, 2017". In the Reference list, data citations must be labeled with "[DATASET]". A data reference must provide the database name, accession number/identifiers and a resolvable link to the landing page from which the data can be accessed at the end of the reference. Further instructions are available at <https://www.embopress.org/page/journal/14693178/authorguide#referencesformat>.

12) As part of the EMBO publication's Transparent Editorial Process, EMBO Reports publishes online a Review Process File to accompany accepted manuscripts. This File will be published in conjunction with your paper and will include the referee reports, your point-by-point response and all pertinent correspondence relating to the manuscript.

Kind regards,

Martina

Referee #1:

Previous studies have shown that mTORC2 can localize to MAMs (ERMCSs) and that its activation in this compartment promotes phosphorylation of Akt and consequently Akt-mediated phosphorylation of MAM proteins such as IP3R and others. In the studies by Kalarikkal and others, they explored the role of the annulate lamellae (AL; a subdomain of the ER) protein Nup358, a nucleoporin, in the ERMCS. First, they found that Nup358 is present in the ERMCS. Depletion of the ERMCS tethering proteins VAPB/PTPIP51, mitofusin 2 or IP3R3 abolished cytoplasmic but not nuclear membrane expression of Nup358 and other AL-associated nucleoporins, suggesting that AL stability at ERMCS depends on the integrity of the latter. By examining the co-expression of VAPB (ER) and PTPIP51 (mitochondria), they found that the depletion of Nup358 enhances contacts/colocalization between the ER and mitochondria in fluorescence, co-IP, proximity ligation (PLA) studies. TEM studies also showed increased mitochondria-ER contacts. These findings indicate that Nup358 negatively regulates ER/mitochondria connectivity (ERMCS). Since the absence of Nup358 enhances ERMCS and given previous findings on the role of mTORC2 in ERMCS, they then analyze the effect of Nup358 depletion on mTORC2 signaling. They found that Nup358 negatively regulates mTORC2-mediated Akt phosphorylation and S6 phosphorylation (mTORC1 signaling). However, the VAPB-PTP1B connectivity is dependent on mTORC2 (rictor) but not mTORC1 (raptor).

The phosphorylation of Akt is specifically dependent on the expression of VAPB-PTPIP51 and not other ERMCS proteins. Furthermore, they showed interaction of the tethering complex with mTORC2 but not mTORC1. This interaction increased in the presence of insulin whereas the interaction with Nup358 was slightly lower. Finally, they showed that the region of Nup358 that includes the Ran-binding domains, kinesin/dynein binding domain and internal repeats was sufficient to interact with VAPB/PTPIP51. Inducible expression of Nup358 was also sufficient to dampen the increased mTORC2/Akt during Nup358 depletion. Based on these findings, they conclude that "Nup358 restricts ER-mitochondria connectivity by suppressing mTORC2/Akt signaling axis"

The studies are interesting and provide new insights on how AL/Nup358 modulates the ERMCS, and that is involved in negative regulation of the interaction of tethering complex/mTORC2 when growth factor signals are low. The data are quite robust and compelling and largely supports their model. The involvement of AL and nucleoporins on the regulation of mTORC2 and ERMCS is intriguing and although the function is currently unknown, the findings here provide new avenues for further studies. There are some issues that should be addressed to strengthen the conclusion.

1. Is the AL-ERMCS interaction stable during absence of growth factors and destabilized by addition of growth factors?
2. In the absence of insulin, the depletion of Nup358 is not sufficient to enhance mTORC2 activation (Fig 3B). This would indicate that PI3K signaling is necessary to localize mTORC2 to the ERMCS. Would Nup358/AL interaction with ERMCS diminish as PI3K signaling is enhanced (and the opposite when PI3K signaling is inhibited?)
3. Verify that the VAPB/PTP151 connectivity is dependent on mTORC2 complex. Knockdown SIN1 and mTOR. Also can treat with Torin vs rapamycin.
4. How does the depletion of VAPB/PTPI51 affect expression of mTORC2 components?
5. There are mislabelings in Fig. 4, Fig 4E, F, G are not included.

6. Since the colocalization of rictor and VAPB was more dramatic than between mTOR and VAPB, they should also examine VAPB and SIN1 to verify that the increased association is due to mTORC2 and not mTORC2-independent rictor.

Referee #2:

Kalarikkal et al have studied the relevance of the annulate lamellae (AL) domain of the ER in ER-mitochondrial contacts. They propose that an interaction of AL with the PTPIP51-VAPB tethering species exerts an inhibition on pAkt-mediated enhancement of the ERMCS abundance. Most of the results linking AL to the contacts is based on targeting of Nup358 (Nup62, Nup88 and Nup214).

The first question is if Nup targeting allows one to conclude on AL. The Authors claim that the lack of Nup358 does not affect AL integrity. Also, they state that Nup384 and Nup214 targeting has opposing effects on the contacts. Furthermore, the different AL resident Nups show opposing patterns of enrichment in the MAM (Fig1B). Based on this, it does not seem to be justified to conclude on AL from the Nup358 phenotypes.

The evidence supporting almost every point is anecdotic, only representative images are shown, and several studies do not meet the standards of current research. Silencing is supposed to be done with 2 different hairpins and the effect on the abundance of the target has to be validated.

I would recommend that the Authors either pursue further studies to link AL to the ERMCS or narrow their statements to Nup358 or Nup214 in terms of ERMCS. In addition, the Authors have to consider the many off target issues with silencing and relying on antibodies and perform the silencing studies with 2 hairpins and validate the effects. Also, co-IP is not a valid approach to demonstrate direct protein-protein interactions. I think these actions are necessary before the study is submitted for publications.

Referee #3:

In this study, the authors demonstrated that annulate lamellae localized nucleoporin Nup358 are present at ER-mitochondria contact sites (ERMCSs). They showed that the reduction of Nup358 increased ERMCSs. They demonstrated that Nup358 interacts with VAPB and PTPIP51. The reduction in Nup358 promotes the interaction between VAPB and PTPIP51, which is abolished by the reduction in the mTORC2 complex component Rictor.

They further showed that loss of Nup358 fosters mTORC2/Akt activity, which is dependent on VAPB and PTPIP51 but not other ERMCS tethers. They found that mTOR and Rictor interact with VAPB and PTPIP51, which is increased upon insulin treatment. They further identified a fragment of Nup358 that interacts with PTPIP51 and VAPB. Ectopic expression of this fragment can suppress mTORC2/Akt activation and reduce the PTPIP51 and VAPB interaction.

The study has some novel points and the experiments were carefully performed. I have a few concerns need to be resolved:

1. The authors emphasized that they found a novel function of annulate lamellae. However, the whole study barely touched this structure. It is reasonable to claim that Nup358 regulates ERMCSs and mTORC2 activity. However, it is not proper to claim that annulate lamellae functions here. One experiment they performed showed that a fragment of Nup358 is sufficient to suppress mTORC2/Akt activation and reduce ERMCSs. There is no high-resolution image to show where the Nup358 truncated form localized. Based on the low-resolution image (Fig. 5F), it seems diffused. If this is true, it excludes the possibility that annulate lamellae play a role in ERMCSs and mTORC2 activity.

2. Nup358 interacts with both PTPIP51 and VAPB, yet it prevents the interaction between PTPIP51 and VAPB. Does that mean that Nup358 forms different subcomplex with these two proteins? Experiments were performed to clarify this issue.

3. What is the functional consequence of loss of Nup358 in cells? less proliferation? Cell death?

4. How does mTORC2 activation increase the interaction between VAPB and PTPIP51? Does it change the levels or modifications of VAPB or PTPIP51?

Referee #4:

In their study, Kalarikkal and colleagues aim to explore the role of Nup358, an annulate lamellae (AL) resident protein, in the

modulation of the ER-mitochondria contact sites (ERMCSs) stability via activation of the mTORC2/Akt pathway. Growing evidence have shown that ERMCSs play important roles in cells, namely by regulating calcium homeostasis, energy metabolism and mitochondrial dynamics.

The main findings are that Nup358 regulates the mTORC2/Akt pathway by modulating the VAPB-PTPIP51 coupling in response to growth factors.

Overall, the study is well conducted, and the obtained results are novel and sound. The manuscript merit further consideration. However, some sets of experiments may be necessary to consolidate the different conclusions.

1. Page 4, last paragraph:

"Moreover, Nup358 interacts with the ERMCS tethering complex VAPB-PTPIP51, and loss of Nup358 leads to enhanced ERMCS stability through increased activation of the mTORC2/Akt pathway." Experiments show an increase in the number of ERMCSs but not in ERMCS stability. Additional experiments would be required to assess this parameter.

"Together, our studies reveal a role for Nup358 in attenuating growth factor mediated stabilization of ERMCSs through a mechanism involving the suppression of mTORC2/Akt activation."

Again, which experiment shows an increase in ERMCSs stabilization? Is there an increased number of ERMCSs in the presence of growth factors?

2. Page 5: "Knockdown of Nup358 led to enhanced contacts between ER and mitochondria, as determined by the increased co-localization between ER and mitochondria in fluorescence microscopy (Fig 1D)."

There is evidence in the field that imply that quantifying the overlap between ER and mitochondrial markers is not the most reliable method to measure the ER-mito contacts. The authors should soften their statement, as here (at this stage of the study, Fig 1D) they only show an increased proximity between the ER and mitochondria.

3. The Figure 1 shows an increase in the ER-mitochondria contacts via VAPB-PTPIP51 in Nup358 deficient cells. What about the other ERMCS tethers?

4. Figure 1F: data are presented as Number of VAPB-PTPIP51 PLA dots / cell. Which marker has been used to assess the cell surface?

5. The experiment depicted in Fig EV1 is not clear. What the authors wanted to demonstrate here? Why staining RanGAP1 and not Nup358? Of note, previous studies showed that knocking down MFN2 leads to an increase in ER-mitochondria association in HeLa cells (<https://doi.org/10.1073/pnas.1504880112>). Is it in line with what authors wanted to show?

6. Fig EV5: Authors should be carefully when using the Mitotracker DeepRed to assess mitochondrial mass. Indeed, the signal of this specific dye can be altered by changes in the mitochondrial membrane potential.

Besides, ER-mitochondria interactions play a role in mitochondrial dynamics. Did the authors observe changes in the shape of the mitochondrial network?

7. Page 8: Figures 4D-G are mentioned in the text but are not found in Figure 4. Please correct the figure labelling.

8. Figure 5F: a PLA signal is visible outside the cells (especially on the images with GFP-Nup358-MC1). Is it because not all the cells were GFP positive? Same question than point 3: Which marker has been used to assess the cell surface? The GFP signal?

9. Method: why different cell types were used for microscopy (U2OS, Huh-7) versus WB (HeLa, HEK293T) experiments.

10. The figure 6 is a bit confusing as it represents what happens in the presence or absence of growth factor (according to the data obtained in the present study). It would have been interesting to also represent what happens in the presence or absence of Nup358.

In the figure legend, the authors write:

"In the presence of growth factors, mTORC2/Akt complex gets recruited to the ERMCSs through interaction with VAPB-PTPIP51 and activated, which phosphorylates and activates Akt. Subsequently, Akt might phosphorylate tethering proteins such as PACS2 (Betz et al., 2013), leading to the stabilization of ERMCSs. In the absence of growth factor signaling, presence of Nup358 at the ERMCSs due to interaction with VAPB and PTPIP51 prevents mTORC2 recruitment and its activation, leading to inhibition of Akt and thereby destabilization of ERMCSs."

It seems that some experimental conditions are missing to support this statement. Is there a change in the number of VAPB-PTPIP51 contacts in the presence vs absence of growth factor? In the presence versus absence of Nup358 (+ or - growth factors) ? It would have been interesting to assess PACS2 phosphorylation by Akt to evaluate ERMCSs stability in Nup358 depleted cells (which would give more strength to the manuscript, supporting the manuscript's title).

11. Nup358 KD increases pAkt: is it a direct cause to effect relationship or can be a consequence of disturbed calcium homeostasis ? A possible gap in the study would be the lack of readout for ERMCSs function in Nup358 depleted cells (e.g. regulation of calcium homeostasis)

Minor :

- all the abbreviations must be explained at the first mention (e.g. VAPB, PTPIP51....)
- Figure 6 is labelled figure 5 in the figure's list.

AUTHORS' RESPONSE TO REVIEWERS' COMMENTS**Referee #1:**

Previous studies have shown that mTORC2 can localize to MAMs (ERMCSs) and that its activation in this compartment promotes phosphorylation of Akt and consequently Akt-mediated phosphorylation of MAM proteins such as IP3R and others. In the studies by Kalarikkal and others, they explored the role of the annulate lamellae (AL; a subdomain of the ER) protein Nup358, a nucleoporin, in the ERMCS. First, they found that Nup358 is present in the ERMCS. Depletion of the ERMCS tethering proteins VAPB/PTPIP51, mitofusin 2 or IP3R3 abolished cytoplasmic but not nuclear membrane expression of Nup358 and other AL-associated nucleoporins, suggesting that AL stability at ERMCS depends on the integrity of the latter. By examining the co-expression of VAPB (ER) and PTPIP51 (mitochondria), they found that the depletion of Nup358 enhances contacts/colocalization between the ER and mitochondria in fluorescence, co-IP, proximity ligation (PLA) studies. TEM studies also showed increased mitochondria-ER contacts. These findings indicate that Nup358 negatively regulates ER/mitochondria connectivity (ERMCS). Since the absence of Nup358 enhances ERMCS and given previous findings on the role of mTORC2 in ERMCS, they then analyze the effect of Nup358 depletion on mTORC2 signaling. They found that Nup358 negatively regulates mTORC2-mediated Akt phosphorylation and S6 phosphorylation (mTORC1 signaling). However, the VAPB-PTP1B connectivity is dependent on mTORC2 (rictor) but not mTORC1 (raptor).

The phosphorylation of Akt is specifically dependent on the expression of VAPB-PTPIP51 and not other ERMCS proteins. Furthermore, they showed interaction of the tethering complex with mTORC2 but not mTORC1. This interaction increased in the presence of insulin whereas the interaction with Nup358 was slightly lower. Finally, they showed that the region of Nup358 that includes the Ran-binding domains, kinesin/dynein binding domain and internal repeats was sufficient to interact with VAPB/PTPIP51. Inducible expression of Nup358 was also sufficient to dampen the increased mTORC2/Akt during Nup358 depletion. Based on these findings, they conclude that "Nup358 restricts ER-mitochondria connectivity by suppressing mTORC2/Akt signaling axis"

The studies are interesting and provide new insights on how AL/Nup358 modulates the ERMCS, and that is involved in negative regulation of the interaction of tethering complex/mTORC2 when growth factor signals are low. The data are quite robust and compelling and largely supports their model. The involvement of AL and nucleoporins on the regulation of mTORC2 and ERMCS is intriguing and although the function is currently unknown, the findings here provide new avenues for further studies. There are some issues that should be addressed to strengthen the conclusion.

Authors' Response: We thank the reviewer for nicely summarizing, highlighting and appreciating our findings. In the revised version, we believe that we have addressed all the concerns raised by the reviewer. Please see the response to individual queries below.

1. Is the AL-ERMCS interaction stable during absence of growth factors and destabilized by addition of growth factors?

Authors' Response: In the first version, AL (Nup358)-VAPB (ERMCS) interactions were shown to be responsive to growth factor (insulin) signalling; specifically, when the growth factor is provided, the AL-ERMCS interaction is significantly reduced as compared to growth factors are absent (Fig. 4C, Previous and Revised versions). Similar results were obtained In the additional experiment performed (Fig. 4D; Revised version).

2. In the absence of insulin, the depletion of Nup358 is not sufficient to enhance mTORC2 activation (Fig 3B). This would indicate that PI3K signaling is necessary to localize mTORC2 to the ERMCS. Would Nup358/AL interaction with ERMCS diminish as PI3K signaling is enhanced (and the opposite when PI3K signaling is inhibited?)

Authors' Response: We thank the reviewer for this critical and insightful concern. We tested if PI3K signalling is required for the insulin-dependent decrease in Nup358-VAPB interaction and included the results in the revised version of the manuscript. As shown in Fig 4D (revised version), inhibition of PI3K activity (with wortmannin treatment), did not affect insulin-dependent reduction in Nup358-VAPB interaction, suggesting that a PI3K-independent pathway may be involved in regulating the AL-ERMCS interaction downstream to growth factor signalling.

3. Verify that the VAPB/PTP151 connectivity is dependent on mTORC2 complex. Knockdown SIN1 and mTOR. Also can treat with Torin vs rapamycin.

Authors' Response: We thank the reviewer for the comments. To confirm that VAPB-PTPIP51 interaction is dependent on mTORC2, we performed an experiment wherein Insulin-dependent stabilization of the VAPB-PTPIP51 interaction was monitored in Rictor or Sin1 depleted cells, and the data is included in the revised version. The results confirmed that insulin-dependent stabilization of VAPB-PTPIP51 interaction is mTORC2 dependent, and not due to an mTORC2-independent function of Rictor (Fig. 3A; Revised version).

4. How does the depletion of VAPB/PTPI51 affect expression of mTORC2 components?

Authors' Response: We monitored the levels of mTORC2 components in VAPB+PTPIP51 depleted cells, and the data is included in the revised version of the manuscript. We do not see any significant change in the protein levels of the mTORC2 components such as mTOR, Rictor, Sin1 and GβL/mLST8 (Appendix Fig. S7; Revised version).

5. There are mislabelings in Fig. 4, Fig 4E, F, G are not included.

Authors' Response: Thank you for pointing this out. We have taken care of this in the revised version.

6. Since the colocalization of rictor and VAPB was more dramatic than between mTOR and VAPB, they should also examine VAPB and SIN1 to verify that the increased association is due to mTORC2 and not mTORC2-independent rictor.

Authors' Response: Thank you very much for the suggestion. We have performed experiments to examine the interaction between VAPB and Sin1 during insulin signalling. The results suggest that similar to VAPB-Rictor interaction, the association between VAPB and Sin1 was also increased upon insulin addition (Fig. 4C Right panel, Revised version).

Referee #2:

Kalarikkal et al have studied the relevance of the annulate lamellae (AL) domain of the ER in ER-mitochondrial contacts. They propose that an interaction of AL with the PTPIP51-VAPB tethering species exerts an inhibition on pAkt -mediated enhancement of the ERMIC abundance. Most of the results linking AL to the contacts is based on targeting of Nup358 (Nup62, Nup88 and Nup214).

The first question is if Nup targeting allows one to conclude on AL. The Authors claim that the lack of Nup358 does not affect AL integrity. Also, they state that Nup384 and Nup214 targeting has opposing effects on the contacts. Furthermore, the different AL resident Nups show opposing patterns of enrichment in the MAM (Fig1B). Based on this, it does not seem to be justified to conclude on AL from the Nup358 phenotypes.

Authors' Response: We thank the reviewer for the comments. We have now restricted our conclusion to Nup358 and toned down our claim on AL.

The evidence supporting almost every point is anecdotic, only representative images are shown, and several studies do not meet the standards of current research. Silencing is supposed to be done with 2 different hairpins and the effect on the abundance of the target has to be validated.

Authors' Response: In the previous version, we had already given evidence for establishing the specificity of Nup358 functions. For example, we had shown that the Nup358's loss of function phenotypes were observed in Nup358 deficient cells generated using two ways of depletion; using siRNA and CRISPR/Cas9-based knockout (KO) (Fig. 2A, B; Previous and Revised versions). Moreover, phenotypes shown by siRNA-mediated depletion of Nup358 was rescued by ectopically expressing an siRNA-resistant version of Nup358 (Fig EV6; Previous & Fig EV3; Revised versions).

I would recommend that the Authors either pursue further studies to link AL to the ERMIC or narrow their statements to Nup358 or Nup214 in terms of ERMIC. In addition, the Authors have to consider the many off target issues with silencing and relying on antibodies and perform the silencing studies with 2 hairpins and validate the effects. Also, co-IP is not a valid approach to demonstrate direct protein-protein interactions. I think these actions are necessary before the study is submitted for publications.

Authors' Response: As per this and other reviewers' comments, we have now restricted our interpretation mostly to Nup358.

The concern regarding the specificity of Nup358's loss of function phenotypes has been addressed in the above response.

With regards to the conclusion on protein-protein interactions, we believe that the reviewer may have got a little confused. Fig. 6C of the previous version represents *in vitro* interaction data, wherein purified MBP-control or MBP-PTPIP51 was mixed with purified GST-Nup358-MC2, and an IP for MBP protein was done. The immunoprecipitates were probed for the presence of GST-Nup358-MC2 with a Nup358 antibody raised against the IR region [as Nup358-MC2 contains the IR region]. Here, co-IP was preferred over direct pull downs with maltose or glutathione

beads, as they were giving strong background in the control pull downs. Co-IP drastically improved the quality of the data as there was very less background in the MBP-control co-IP sample (Fig. 6C, Previous and Revised version).

To increase the clarity and to avoid confusion, we have included a label “*In vitro* interaction studies using purified proteins” below the data (Fig 6C, Revised version)

Referee #3:

In this study, the authors demonstrated that annulate lamellae localized nucleoporin Nup358 are present at ER-mitochondria contact sites (ERMCSs). They showed that the reduction of Nup358 increased ERMCSs. They demonstrated that Nup358 interacts with VAPB and PTPIP51. The reduction in Nup358 promotes the interaction between VAPB and PTPIP51, which is abolished by the reduction in the mTORC2 complex component Rictor.

They further showed that loss of Nup358 fosters mTORC2/Akt activity, which is dependent on VAPB and PTPIP51 but not other ERMCS tethers. They found that mTOR and Rictor interact with VAPB and PTPIP51, which is increased upon insulin treatment.

They further identified a fragment of Nup358 that interacts with PTPIP51 and VAPB. Ectopic expression of this fragment can suppress mTORC2/Akt activation and reduce the PTPIP51 and VAPB interaction.

The study has some novel points and the experiments were carefully performed. I have a few concerns need to be resolved:

Authors' Response: We thank the reviewer for the positive comments and appreciating the work. Specific response to each query is given below.

1. The authors emphasized that they found a novel function of annulate lamellae. However, the whole study barely touched this structure. It is reasonable to claim that Nup358 regulates ERMCSs and mTORC2 activity. However, it is not proper to claim that annulate lamellae functions here. One experiment they performed showed that a fragment of Nup358 is sufficient to suppress mTORC2/Akt activation and reduce ERMCSs. There is no high-resolution image to show where the Nup358 truncated form localized. Based on the low-resolution image (Fig. 5F), it seems diffused. If this is true, it excludes the possibility that annulate lamellae play a role in ERMCSs and mTORC2 activity.

Authors' Response: We thank the reviewer for this critical concern that we agree with. In the revised manuscript, we have emphasized the role of Nup358 in the processes studied in the paper and downplayed the claims on AL.

2. Nup358 interacts with both PTPIP51 and VAPB, yet it prevents the interaction between PTPIP51 and VAPB. Does that mean that Nup358 forms different subcomplex with these two proteins? Experiments were performed to clarify this issue.

Authors' Response: We appreciate the critical concern raised by the reviewer. At present we do not have any indications on the mechanistic details of how Nup358 can bind to both VAPB

and PTPIP51 and abrogate inter-molecular interaction between VAPB and PTPIP51. As mentioned by the reviewer, Nup358 might make different subcomplexes. A few possibilities have been included in the 'Discussion' section (2nd para under 'Discussion'; Page 11; highlighted in Red).

3. What is the functional consequence of loss of Nup358 in cells? less proliferation? Cell death?

Effect of Nup358 depletion on cell proliferation and cell death was monitored. There was no significant effect on either of the parameters. See below under Reviewer's Fig. 1.

Reviewer's Fig.1. Cell cycle analysis and cell death (Apoptosis) analysis in Nup358 depleted condition

4. How does mTORC2 activation increase the interaction between VAPB and PTPIP51? Does it change the levels or modifications of VAPB or PTPIP51?

Authors' Response: We thank the reviewer for the comment. Previously it has been shown that GSK3 β can destabilize VAPB-PTPIP51 interactions (1, 2). Our new data (Fig. 5, Revised version) suggest that mTORC2 mediated inhibitory phosphorylation of GSK3 β may result in the increased VAPB-PTPIP51 interaction observed with mTORC2 activation.

Referee #4:

In their study, Kalarikkal and colleagues aim to explore the role of Nup358, an annulate lamellae (AL) resident protein, in the modulation of the ER-mitochondria contact sites (ERMCSs) stability via activation of the mTORC2/Akt pathway. Growing evidence have shown that ERMCSs play

important roles in cells, namely by regulating calcium homeostasis, energy metabolism and mitochondrial dynamics.

The main findings are that Nup358 regulates the mTORC2/Akt pathway by modulating the VAPB-PTPIP51 coupling in response to growth factors.

Overall, the study is well conducted, and the obtained results are novel and sound. The manuscript merit further consideration. However, some sets of experiments may be necessary to consolidate the different conclusions.

Authors' Response: We thank the reviewer for appreciating the work and supporting the manuscript. Specific response to the Reviewer's queries has been given below.

1. Page 4, last paragraph:

"Moreover, Nup358 interacts with the ERMCS tethering complex VAPB-PTPIP51, and loss of Nup358 leads to enhanced ERMCS stability through increased activation of the mTORC2/Akt pathway." Experiments show an increase in the number of ERMCSs but not in ERMCS stability. Additional experiments would be required to assess this parameter.

"Together, our studies reveal a role for Nup358 in attenuating growth factor mediated stabilization of ERMCSs through a mechanism involving the suppression of mTORC2/Akt activation."

Again, which experiment shows an increase in ERMCSs stabilization? Is there an increased number of ERMCSs in the presence of growth factors?

Authors' Response: We thank the reviewer for the comments. In general, we have resorted to the usage of 'increased ER-mitochondria connectivity' instead of 'ERMCS stability'.

We have performed additional experiments to show that insulin increases the ER-mitochondria connectivity, and included the data in the revised version of this manuscript. The results suggest that growth factor (insulin) treatment enhances the interaction between VAPB and PTPIP51, an indicator of ERMCSs (Fig. 3A). Furthermore, we show that the insulin-dependent increase in ER-mitochondria connectivity depends on mTORC2, as depletion of Rictor or Sin1 (mTORC2-specific subunits) abrogated this effect of insulin (Fig. 3A; Revised version).

2. Page 5: "Knockdown of Nup358 led to enhanced contacts between ER and mitochondria, as determined by the increased co-localization between ER and mitochondria in fluorescence microscopy (Fig 1D)."

There is evidence in the field that imply that quantifying the overlap between ER and mitochondrial markers is not the most reliable method to measure the ER-mito contacts. The authors should soften their statement, as here (at this stage of the study, Fig 1D) they only show an increased proximity between the ER and mitochondria.

Authors' Response: We thank the reviewer for pointing this out. We have rephrased the sentences to reflect this (Page 6, highlighted in red).

3. The Figure 1 shows an increase in the ER-mitochondria contacts via VAPB-PTPIP51 in Nup358 deficient cells. What about the other ERMCS tethers?

Authors' Response: We thank the reviewer for this suggestion. We have performed the experiments and included the data showing an increase in the interaction between BAP31 and Fis1 (another ER-mitochondria tether complex), in addition to VAPB-PTPIP51, when Nup358 is depleted (Fig. 1G; Revised version).

4. Figure 1F: data are presented as Number of VAPB-PTPIP51 PLA dots / cell. Which marker has been used to assess the cell surface?

Authors' Response: We thank the reviewer for the query. We have used phase contrast images to demarcate the cell periphery, and the CellProfiler software to quantitate the number of PLA dots per cell. This information is included in the 'Methods and Materials' section (Page 19, highlighted in Red).

5. The experiment depicted in Fig EV1 is not clear. What the authors wanted to demonstrate here? Why staining RanGAP1 and not Nup358? Of note, previous studies showed that that knocking down MFN2 leads to an increase in ER-mitochondria association in HeLa cells (<https://doi.org/10.1073/pnas.1504880112>). Is it in line with what authors wanted to show?

Authors' Response: We thank the reviewer for pointing this out. RanGAP1 is a strong interacting partner of Nup358 and has been shown to co-localize with Nup358 at the nuclear envelope and AL. A sentence to clarify this point is added in the Figure legends (Figure EV1; highlighted in Red).

6. Fig EV5: Authors should be carefully when using the Mitotracker DeepRed to assess mitochondrial mass. Indeed, the signal of this specific dye can be altered by changes in the mitochondrial membrane potential.

Authors' Response: We thank the reviewer for this valuable suggestion. We have performed the experiment using MitoTracker Green dye and included the data in the revised version (Appendix Fig. S4 in the Revised version; highlighted in Fig legends in Red).

Besides, ER-mitochondria interactions play a role in mitochondrial dynamics. Did the authors observe changes in the shape of the mitochondrial network?

We have been currently analysing this aspect and it may become a part of a future study. We believe it is beyond the scope of this manuscript.

7. Page 8: Figures 4D-G are mentioned in the text but are not found in Figure 4. Please correct the figure labelling.

Authors' Response: We thank the reviewer for pointing this out. We made the corrections in the revised version.

8. Figure 5F: a PLA signal is visible outside the cells (especially on the images with GFP-Nup358-MC1). Is it because not all the cells were GFP positive? Same question than point 3: Which marker has been used to assess the cell surface? The GFP signal?

Authors' Response: We thank the reviewer for this query. In fact even though it was an inducible stable line, some cells expressed no or low GFP, which was below the microscopic

detection limit. Therefore we have quantitated the PLA dots only in the visually obvious GFP-positive cells.

To improve the visual impact and appreciate the difference the cell periphery has been shown with dotted lines in Figure 5F (Fig. 6F; Revised version) and in a new data added (Fig. 5B, Revised version).

9. Method: why different cell types were used for microscopy (U2OS, Huh-7) versus WB (HeLa, HEK293T) experiments.

Authors' Response: We thank the reviewer for this query. Using different cell lines ensured that the new findings are not cell type specific, but are of a general nature.

Moreover, the cell lines were also chosen based on the experiments taking into consideration the advantage of each cell lines: HeLa cells were used for most of the experiments; HEK293T cells were often used when ectopic expression of proteins was required; U2OS cells have flat and thin morphology, and therefore, was used for better detailed imaging of the subcellular structures; Huh-7 showed a higher density of AL and was used for observing AL-mitochondria associations by Super resolution microscopy.

10. The figure 6 is a bit confusing as it represents what happens in the presence or absence of growth factor (according to the data obtained in the present study). It would have been interesting to also represent what happens in the presence or absence of Nup358.

Authors' Response: We thank the reviewer for the suggestion. We have performed new experiments to see VAPB-PTPIP51 (ERMCSs) in the presence and absence of insulin in control and Nup358 depleted cells. The new data is included in the revised version (Fig. 5D). The results show that in control cells, the VAPB-PTPIP51 interaction significantly increased upon Insulin addition. Interestingly, in Nup358-depleted cells, the VAPB-PTPIP51 interaction was already at a maximum even in the absence of insulin, and insulin addition did not further increase the interaction between VAPB and PTPIP51.

In the figure legend, the authors write:

"In the presence of growth factors, mTORC2/Akt complex gets recruited to the ERMCSs through interaction with VAPB-PTPIP51 and activated, which phosphorylates and activates Akt. Subsequently, Akt might phosphorylate tethering proteins such as PACS2 (Betz et al., 2013), leading to the stabilization of ERMCSs. In the absence of growth factor signaling, presence of Nup358 at the ERMCSs due to interaction with VAPB and PTPIP51 prevents mTORC2 recruitment and its activation, leading to inhibition of Akt and thereby destabilization of ERMCSs."

Authors' Response: We thank the reviewer for the comments. In this revised version, we have new results suggesting that GSK3 β could be playing a crucial role downstream to Nup358/mTORC2 (Fig 5; Revised version). We show that ectopic expression of GSK3 β negates the effect of Nup358 depletion on the ERMCSs. This could also mean that downstream to Nup358, GSK3 β mediated disruption of VAPB-PTPIP51 interaction could majorly contribute to the destabilization of ERMCSs and mTORC2/Akt could inhibit GSK3 β to achieve stabilization of ERMCSs downstream to growth factor signalling.

We also find that Nup358 interacts with GSK3 β (Fig. 5C, Revised manuscript). However, the mechanistic details of how the interplay between mTORC2, VAPB-PTPIP51 complex, Nup358 and GSK3 β is contributing to the remodelling of ERMCS requires further research (included in 'Discussion' section, pages 12 & 13).

It seems that some experimental conditions are missing to support this statement. Is there a change in the number of VAPB-PTPIP51 contacts in the presence vs absence of growth factor? In the presence versus absence of Nup358 (+ or - growth factors) ?

We appreciate the reviewer's concern regarding the same. This issue has been addressed in the revised manuscript Fig. 5D, as described above (Response to #10).

It would have been interesting to assess PACS2 phosphorylation by Akt to evaluate ERMCSs stability in Nup358 depleted cells (which would give more strength to the manuscript, supporting the manuscript's title).

Authors' Response: We thank the reviewer for the comments. Based on the new data, it appears that mTORC2/Akt mediated inhibition of GSK3 β , majorly contributes to the stabilization of ERMCS downstream to Nup358 & growth factor signalling. Particularly, under Nup358 depleted condition, expression of GSK3 β almost completely rescued the increased contacts between ER and mitochondria (Fig. 5B; Revised version).

11. Nup358 KD increases pAkt: is it a direct cause to effect relationship or can be a consequence of disturbed calcium homeostasis? A possible gap in the study would be the lack of readout for ERMCSs function in Nup358 depleted cells (e.g. regulation of calcium homeostasis)

Authors' Response: We thank the reviewer for the comments. In Nup358 deficient cells, we expect that the ERMCS functions will be affected, and we are further exploring in this direction. We believe it is beyond the scope of this manuscript.

Minor :

- all the abbreviations must be explained at the first mention (e.g. VAPB, PTPIP51....)

We thank the Reviewer for this suggestion. We have made changes accordingly (Page 3, highlighted in Red).

- Figure 6 is labelled figure 5 in the figure's list.

We appreciate the Reviewer for pointing this. We made corrections in the Revised version.

References

1. Stoica,R., Paillusson,S., Gomez-Suaga,P., Mitchell,J.C., Lau,D.H., Gray,E.H., Sancho,R.M., Vizcay-Barrena,G., De Vos,K.J., Shaw,C.E., *et al.* (2016) ALS / FTD -associated FUS activates GSK3 β to disrupt the VAPB – PTPIP 51 interaction and ER –mitochondria

associations. *EMBO Rep.*, **17**, 1326–1342.

2. Stoica,R., De Vos,K.J., Paillusson,S., Mueller,S., Sancho,R.M., Lau,K.F., Vizcay-Barrena,G., Lin,W.L., Xu,Y.F., Lewis,J., *et al.* (2014) ER-mitochondria associations are regulated by the VAPB-PTPIP51 interaction and are disrupted by ALS/FTD-associated TDP-43. *Nat. Commun.*, **5**, 3996.

Dear Joe,

Thank you once more for the submission of your revised manuscript to EMBO reports. I had already informed you about the positive feedback from the referees (copied again below).

I have now completed the checks from the editorial side and there are a few things that we need before we can proceed with the official acceptance of your study.

- Please update the 'Conflict of interest' paragraph to our new 'Disclosure and competing interests statement'. For more information see

<https://www.embopress.org/page/journal/14693178/authorguide#conflictsofinterest>

- Regarding the Author Contributions, we now use CRediT to specify the contributions of each author in the journal submission system. Therefore, please remove the Author Contributions from the manuscript file and make sure that the author contributions in our manuscript tracking system are correct and up-to-date. The information you specified in the system will be automatically retrieved and typeset into the article. You can enter additional information in the free text box provided, if you wish.

- The information on funding needs to be completed in the online manuscript tracking system and the information must match that given in the Acknowledgments paragraph. Currently missing from the system: the Science and Engineering Research Board (SERB), intramural funding from NCCS, Financial support from the Council of Scientific and Industrial Research, Ministry of Science and Technology, University Grants Commission (UGC).

The funder in the Comments textbox needs to be removed from the textbox and provided as a separate funder.

NIH P40OD018537 is acknowledged for obtained stocks so this also needs to be entered in the manuscript tracking system.

- Please add page numbers to the Appendix and to its table of content.

- Figure 7: some of the labels and shapes in the model figure are not well visible, they appear only as shades and outlines. This might be a PDF compression artefact. Please check this figure and resupply it.

- Author Checklist: Please add the information on Mycoplasma testing also to the Methods section, where relevant.

- Source data: The folder for Figure 3 needs to be corrected as it currently contains source data for ALL figures (not just Figure 3).

- Our production/data editors have asked you to clarify several points in the figure legends (see below). Please incorporate these changes in the manuscript and return the revised file with tracked changes with your final manuscript submission.

A) Statistical test information. Only p-values that are actually shown in the figure panel(s) should (and must) be defined in the legends, all others should be removed from (or added to) the legend. Moreover, we ask for the specification of exact p-values:

- Please note that the exact p-values are not provided in the legends of figures 1d-h; 2a-f; 3a-c; 4c-d; 5a-b, d; 6c-f; EV 2a-b; EV 3; EV 4c; EV 5b.

- Please note that in figures 1g; 6e; there is a mismatch between the annotated p values in the figure legend and the annotated p values in the figure file that should be corrected.

- Although 'n' is provided, please describe the nature of entity for 'n' in the legend of figure EV 3.

B) Replicates, error bars, and data presentation:

- Please note that the scale bar needs to be defined for figure 4d.

- Please note that the asterisk and arrows are not defined in the legend of figure 1e. This needs to be rectified.

- As a standard procedure, we edit the Title and Abstract to make it more accessible to a general readership. Here, I have only corrected some typos in the Abstract but suggest to change the title so as to avoid the term 'axis'. Please see my suggestion below my signature.

- Finally, EMBO Reports papers are accompanied online by

A) a short (1-2 sentences) summary of the findings and their significance,

B) 2-3 bullet points highlighting key results and

C) a schematic summary figure that provides a sketch of the major findings (not a data image).

Please provide the summary figure as a separate file in PNG or JPG format at a size of 550x300-600 pixels (width x height).

Please note that the size is rather small and that text needs to be readable at the final size. Please send us this information along with the revised manuscript.

- On a different note, I would like to alert you that EMBO Press offers a new format for a video-synopsis of work published with us, which essentially is a short, author-generated film explaining the core findings in hand drawings, and, as we believe, can be very useful to increase visibility of the work. This has proven to offer a nice opportunity for exposure i.p. for the first author(s) of the study. Please see the following link for representative examples and their integration into the article web page:

<https://www.embopress.org/doi/full/10.15252/emboj.2019103932>

With kind regards,

Martina

Referee #1:

In this revision, the authors have conducted more experiments to support their conclusion. In particular, they strengthened the conclusion that that Nup358 regulates mTORC2 signaling. The additional data on the PI3K-independence of this regulation is intriguing. The authors have satisfactorily addressed my comments.

Referee #3:

In the revised manuscript, the authors addressed my concerns about the annulate lamellae and focused on Nup358's function. They also performed some experiments to resolve my other concerns. I think the current version of manuscript is suitable for publishing on EMBO Reports.

Referee #4:

In the revised version of their manuscript, the authors have performed additional experiments strengthening their conclusions and improving the quality of the manuscript. They have addressed all of my concerns, and I believe that the manuscript is now suitable for publication.

Nup358 restricts ER-mitochondria connectivity by suppressing mTORC2/Akt signaling

ER-mitochondria contact sites (ERMCSs) regulate processes including calcium homeostasis, energy metabolism and autophagy. Previously, it was shown that during growth factor signalling, mTORC2/Akt gets recruited to and stabilizes ERMCSs. Independent studies showed that GSK3 β , a well-known Akt substrate, reduces ER-mitochondria connectivity by disrupting the VAPB-PTPIP51 tethering complex. However, the mechanisms that regulate ERMCSs are incompletely understood. Here we find that annulate lamellae (AL), relatively unexplored subdomains of ER enriched with a subset of nucleoporins, are present at ERMCSs. Depletion of Nup358, an AL-resident nucleoporin, results in enhanced mTORC2/Akt activation, GSK3 β inhibition and increased ERMCSs. Depletion of Rictor, an mTORC2-specific subunit or exogenous expression of GSK3 β was sufficient to reverse the ERMCS-phenotype in Nup358-deficient cells. We show that growth factor-mediated activation of mTORC2 requires the VAPB-PTPIP51 complex, whereas, Nup358's association with this tether restricts mTORC2/Akt signalling and ER-mitochondria connectivity. Expression of a Nup358 fragment that is sufficient for interaction with the VAPB-PTPIP51 complex suppresses mTORC2/Akt activation and disrupts the ERMCSs. Collectively, our study uncovers a novel role for Nup358 in controlling ERMCSs by modulating the mTORC2/Akt/GSK3 β axis.-

All editorial and formatting issues were resolved by the authors.

Dr. Jomon Joseph
National Centre for Cell Science
Cell Biology
Ganeshkhind
Pune, Maharashtra 411 007
India

Dear Joe,

Thank you for submitting your revised manuscript. I have completed all editorial checks and all looks fine. I am therefore very pleased to accept your manuscript for publication in the next available issue of EMBO reports. Thank you for your contribution to our journal and congratulations on a nice work!

Best regards,

Martina
